# A Joint Diffusion Model with Pre-Trained Priors for RNA Sequence-Structure Co-Design

**Xiner Li**[1]   **Masatoshi Uehara**[2]   **Xingyu Su**[1]   **Gabriele Scalia**[3]   **Shuiwang Ji**[1*]
[1]Texas A&M University    [2]Chan Zuckerberg Initiative    [3]Genentech, Inc.

## Abstract

RNA molecules underlie regulation, catalysis, and therapeutics in biological systems, yet de novo RNA design remains difficult. The RNA sequence-structure co-design problem generates nucleotide sequences and 3D conformations jointly, which is challenging due to RNA's conformational flexibility, non-canonical base pairing, and the scarcity of experimentally resolved 3D data. We introduce a joint generative framework that embeds RoseTTAFold2NA as the denoiser into a dual diffusion model, injecting rich cross-molecular priors while enabling sample-efficient learning from limited RNA data. Our method couples a discrete diffusion process for sequences with an $SE(3)$-equivariant diffusion for rigid-frame translations and rotations over all-atom coordinates. The architecture supports flexible conditioning, and is further enhanced at inference via lightweight RL techniques that optimize task-aligned rewards. Across de novo RNA design as well as complex and protein-conditioned design tasks, our approach yields high self-consistency and confidence scores, improving over recent diffusion/flow baselines trained from scratch. Results demonstrate that leveraging pre-trained structural priors within a joint diffusion framework is a powerful paradigm for RNA design under data scarcity, enabling high-fidelity generation of standalone RNAs and functional RNA-protein interfaces.

## 1 Introduction

Ribonucleic acid (RNA) molecules play fundamental roles in cellular processes, from catalysis and gene regulation to protein synthesis and viral replication (Conesa et al., 2016; Miao & Westhof, 2017; Keefe et al., 2010). The ability to computationally design RNA sequences that fold into specific three-dimensional (3D) structures holds immense promise for synthetic biology, therapeutic development, and nanotechnology applications (Li et al., 2026). However, RNA design remains significantly more challenging than protein design due to the severe scarcity of experimentally determined RNA structures, creating a fundamental bottleneck for data-driven methods (Wang et al., 2025; Zhang et al., 2024). This disparity necessitates novel methodologies that can effectively leverage limited structural data while capitalizing on the broader knowledge encoded in sequence information and cross-molecular interactions (Zhang et al., 2025).

Diffusion models (Sohl-Dickstein et al., 2015; Ho et al., 2020; Song et al., 2020) have emerged as the state-of-the-art approach for generation tasks under data-scarce conditions (Prabhudesai et al., 2025). This advantage stems from their ability to learn robust distributions through iterative refinement, making them more sample-efficient and less prone to overfitting on small datasets. Recent theoretical and empirical analyses have shown that diffusion models can achieve better coverage of the data distribution with fewer samples, as they learn to denoise from multiple corruption levels simultaneously (Prabhudesai et al., 2025). The RNA community has begun to recognize the potential of diffusion-based approaches, with several recent works showing promise with score-based models and flow matching techniques (Gao & Lu, 2024; Huang et al., 2024a; Rubin et al., 2025; Morehead et al., 2023). However, these approaches primarily focus on training diffusion models from scratch on RNA-specific datasets, limiting their performance due to data constraints.

---

*Correspondence to Shuiwang Ji <sji@tamu.edu>

Recent advances in protein design have demonstrated the transformative potential of extending pre-trained structure prediction models into generative frameworks. The development of RFdiffusion (Watson et al., 2022), which builds on RoseTTAFold (Baek et al., 2023), exemplifies this paradigm: a structure prediction model trained on large-scale protein data was successfully adapted into a diffusion-based generative model capable of designing novel protein structures. This approach elegantly sidesteps the data scarcity problem by leveraging the rich representational knowledge learned during discriminative pre-training, then refining it through a generative process that explores the manifold of physically plausible structures (Lisanza et al., 2023; 2024). This naturally raises the question of whether similar principles can be applied to RNA design, where data limitations are even more severe. Recently, beyond RoseTTAFold (Baek et al., 2023), which is limited to the protein domain, RoseTTAFold2NA (Baek et al., 2022) marked a significant advancement in biomolecular structure prediction by unifying the modeling of proteins, RNA, DNA, and their complexes within a single architectural framework. This comprehensive pre-training provides an exceptional foundation for generative modeling, as the model has already learned to recognize and encode the fundamental principles governing RNA structure formation, protein-RNA recognition, and the intricate interplay between sequence and structure across molecular types. No existing work has explored the integration of a comprehensively pre-trained biomolecular model into a diffusion framework for RNA design. This gap represents a significant missed opportunity, as the combination of rich pre-trained representations with the generative power of diffusion models could overcome the fundamental limitations that have historically constrained RNA design.

In this work, we target the problem of RNA sequence-structure co-design. Our work introduces RiboDiff, the first diffusion-based generative framework built upon a pre-trained biomolecular model targeting RNA joint generation. By embedding RoseTTAFold2NA within a carefully designed diffusion process, we enable the joint generation of RNA sequences and their corresponding 3D structures through a single coherent framework. This approach offers several key advantages over existing methods. First, it leverages the extensive cross-domain knowledge encoded in RoseTTAFold2NA, including understanding of protein-RNA interactions, structural motifs, and sequence-structure relationships learned from diverse molecular contexts. Second, the diffusion framework provides a principled approach to exploring the space of possible RNA designs while maintaining physical plausibility through the learned priors. Third, the joint modeling of sequence and structure addresses the fundamental coupling between these modalities, avoiding the limitations of two-stage approaches that design one before the other (Dotu et al., 2014; Joshi et al., 2025; Anand et al., 2025).

Specifically, we implement discrete diffusion for categorical nucleotide sequences coupled with SE(3)-equivariant continuous diffusion for three-dimensional structures, ensuring that the generative process respects both the discrete nature of sequence space and the geometric constraints of molecular structures. Furthermore, we introduce the use of inference-time reinforcement learning (RL) enhancement (Li et al., 2024b; Uehara et al., 2025), which leverages the model's own predictive capabilities and diffusion process to guide the generation process toward high-quality designs. The framework naturally extends to conditional generation tasks with the cross-molecular knowledge encoded in pre-trained priors, enabling the design of RNA molecules that interact with specific protein targets. This capability addresses critical applications in therapeutic design, where RNA aptamers, regulatory elements, or catalytic RNAs must be engineered to bind predetermined protein partners with high affinity and specificity. Through experiments on de novo design for solo RNA, RNA-protein complex, and protein-conditioned RNA tasks, we demonstrate that our approach substantially outperforms existing methods, achieving state-of-the-art results in a computationally effective way. These results validate the principle that extending pre-trained models into diffusion frameworks represents a powerful paradigm for biomolecular design, particularly in data-limited domains like RNA structure. We emphasize our novel contributions as (1) the first RF2NA-based joint sequence-structure diffusion model for RNA, with (2) explicit discrete-continuous co-diffusion, and (3) conditional RNA-protein co-design with RL-style inference.

## 2 RELATED WORKS

**Inverse RNA Folding.** Recently, learning-based methods have been studied for 3D inverse design. gRNAde (Joshi et al., 2025) introduced an SE(3)-equivariant graph neural network that generates RNA sequences conditioned on a fixed 3D backbone, analogous to how ProteinMPNN designs protein sequences for a given fold. Ribodiffusion (Huang et al., 2024a) performs RNA inverse

folding with a diffusion model consisting of a graph neural network-based structure module and a Transformer-based sequence module. However, inverse design methods address only the sequence optimization aspect, requiring a predefined structure as input instead of generating RNA shapes de novo.

**Joint RNA Generation.** Recent works have aimed to co-generate RNA sequences and structures together without a fixed template. MMDiff (Morehead et al., 2023) pioneered a diffusion-based approach for joint sequence-structure modeling across biomolecules with a DDPM, demonstrating the feasibility of simultaneously generating nucleic acids and protein structures along with sequences. However, as a multi-domain model trained from scratch on limited RNA data, its RNA designs were of modest accuracy. RiboGen (Rubin et al., 2025) simultaneously generate full RNA sequences and all-atom 3D structures via flow matching using a Euclidean-equivariant neural network with coupled continuous and discrete flow. An alternative strategy is to break the design problem into stages, generating a backbone structure first and then optimizing a sequence for it, which RNA-FrameFlow (Anand et al., 2025) employs using $SE(3)$-flow matching. However, FrameFlow does not produce sequences, but relies on external inverse folding using gRNAde to assign a sequence to each generated backbone. This two-step design may miss global sequence-structure optimality since the sequence is not co-optimized during structure generation, while joint generation avoids the need of post-hoc external tools. Both MMDiff and RiboGen confirm that joint generative modeling of RNA is possible; yet, they rely on training bespoke models on the scarce RNA structure data. In contrast, our approach leverages a pre-trained multi-context model within a diffusion framework, which injects extensive prior knowledge and improves sample efficiency and fidelity.

**Conditional RNA Design.** Beyond de novo RNA design, several methods condition generation on binding partners or other context. RNAFlow (Nori & Jin, 2024) targets protein-RNA complex design conditioning on a given protein structure, and uses a GNN to propose RNA sequences and employs RoseTTAFold2NA to predict the RNA's 3D backbone. Building on this, RNA-EFM (Abir & Zhang, 2025) incorporated physics-based priors into a flow-matching model for protein-conditioned RNA design. RiboFlow (Ma et al., 2025) extends conditional design to small-molecule targets, introducing a ligand-conditioned flow-matching model that co-designs RNA sequences and structures with a specified small-molecule bound in the RNA's pocket. These conditional generative models demonstrate the growing need to target RNA designs for specific functions, but they generally require complex pipelines with external tools or extensive data collection, while our framework produces sequence and structure together in one distribution, either unconditionally or with optional conditions.

## 3 PRELIMINARIES

### 3.1 PROBLEM FORMULATION

The RNA sequence-structure co-design problem seeks to simultaneously generate RNA sequences and their corresponding 3D structures that satisfy specific functional requirements. Unlike some traditional approaches that treat sequence design and structure prediction as separate tasks, here we formulate this as a joint generative problem over the coupled sequence-structure space.

Formally, we define the RNA co-design problem as learning a joint distribution $p(\mathbf{s}, \mathbf{X})$, where $\mathbf{s} \in \mathcal{S}^L$ represents an RNA sequence of length $L$ from the discrete alphabet $\mathcal{S} = \{A, C, G, U, N\}$, and $\mathbf{X} \in \mathbb{R}^{L \times N_a \times 3}$ denotes the three-dimensional (3D) coordinates of all atoms in the molecule, with $N_a$ atoms per nucleotide. The challenge lies in capturing the complex bidirectional dependencies between sequence and structure: sequences determine folding patterns through base pairing and stacking interactions, while structural constraints restrict viable sequence spaces. The co-design objective can be expressed as an optimization problem

$$\mathbf{s}^*, \mathbf{X}^* = \arg \max_{(\mathbf{s}, \mathbf{X}) \in \mathcal{V}} p(\mathbf{s}, \mathbf{X}) \cdot f_{\text{objective}}(\mathbf{s}, \mathbf{X}), \tag{1}$$

where $\mathcal{V}$ represents the set of valid sequence-structure pairs satisfying physical constraints, and $f_{\text{objective}}$ encodes design objective of self-consistency between $\mathbf{s}$ and $\mathbf{X}$, while other properties such as diversity, binding affinity, or catalytic activity can also be included. With the generative modeling formulation, we learn to sample from the joint distribution $p(\mathbf{s}, \mathbf{X})$ directly, which enables flexible conditioning for various design scenarios. For de novo design we sample from the unconditional distribution $p(\mathbf{s}, \mathbf{X})$. For therapeutic applications, we can condition on protein binding partners $p(\mathbf{s}^{\text{RNA}}, \mathbf{X}^{\text{RNA}} | \mathbf{X}^{\text{protein}}, \mathbf{s}^{\text{protein}})$ to design RNA molecules with specific interaction properties.

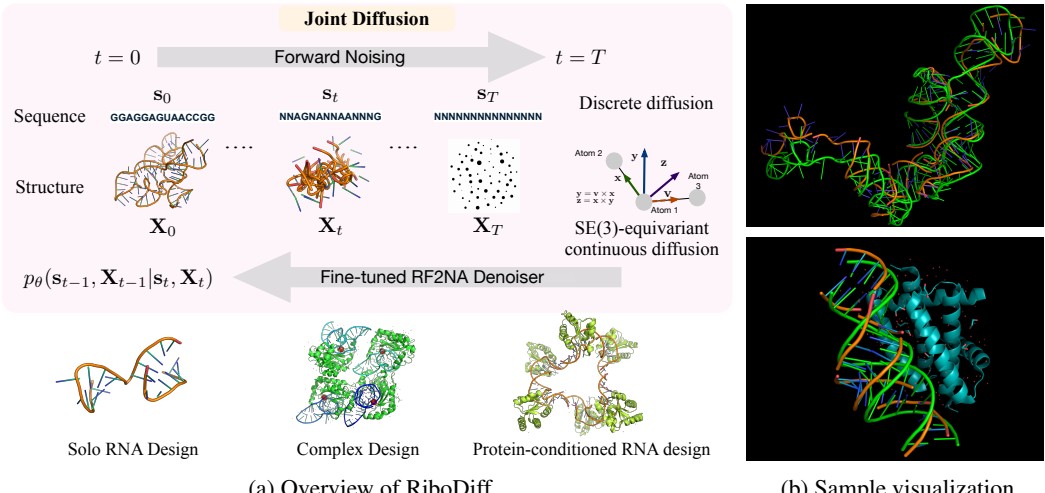

(a) Overview of RiboDiff                    (b) Sample visualization

Figure 1: **(a)** Joint diffusion with pre-trained priors for RNA co-design. Starting from clean sequence–structure, the forward process applies discrete diffusion to nucleotides and $SE(3)$-equivariant diffusion to all-atom frames. We fine-tune the pre-trained RF2NA as the denoiser, enabling joint generation and flexible conditioning across multiple tasks. **(b)** Orange traces show co-designed RNAs while green traces are reference RNAs; cyan ribbons denote protein in conditional settings.

## 3.2 DIFFUSION MODELS

Diffusion models have been widely applied in generative tasks across various fields. Denoising diffusion probabilistic models (DDPMs) define a forward noising process that gradually corrupts data $\mathbf{x}_0 \sim q(\mathbf{x}_0)$ into noise over $T$ timesteps, and learn a reverse denoising process to generate samples from noise. The forward process is defined as a Markov chain $q(\mathbf{x}_{1:T}|\mathbf{x}_0) = \prod_{t=1}^{T} q(\mathbf{x}_t|\mathbf{x}_{t-1})$. For continuous data, the forward transitions follow $q(\mathbf{x}_t|\mathbf{x}_{t-1}) = \mathcal{N}(\mathbf{x}_t; \sqrt{1-\beta_t}\mathbf{x}_{t-1}, \beta_t\mathbf{I})$ with noise schedule $\{\beta_t\}_{t=1}^{T}$. This admits a closed-form marginal $q(\mathbf{x}_t|\mathbf{x}_0) = \mathcal{N}(\mathbf{x}_t; \sqrt{\bar{\alpha}_t}\mathbf{x}_0, (1-\bar{\alpha}_t)\mathbf{I})$, where $\alpha_t = 1 - \beta_t$ and $\bar{\alpha}_t = \prod_{s=1}^{t} \alpha_s$.

The reverse process learns to denoise by parameterizing $p_\theta(\mathbf{x}_{t-1}|\mathbf{x}_t)$, typically through predicting the noise $\boldsymbol{\epsilon}$ or the clean data $\mathbf{x}_0$. The training objective minimizes the variational lower bound, which simplifies to $\mathcal{L}_{\text{simple}} = \mathbb{E}_{t,\mathbf{x}_0,\boldsymbol{\epsilon}}\left[\|\boldsymbol{\epsilon} - \boldsymbol{\epsilon}_\theta(\mathbf{x}_t, t)\|^2\right]$.

## 4 RIBODIFF: JOINT DIFFUSION MODEL FOR RNA CO-DESIGN

We propose a unified diffusion-based framework for RNA sequence-structure co-design that jointly models a discrete nucleotide sequence and an all-atom 3D conformation. The key idea is to embed a powerful pretrained biomolecular structure prediction model, RoseTTAFold2NA (RF2NA), as the denoising network inside a dual diffusion process, i.e., a discrete diffusion on the categorical sequence space and an $SE(3)$-equivariant diffusion on the space of rigid frames that reconstruct all-atom coordinates. This construction lets us respect the heterogeneous nature of sequence and structure, leverage cross-molecular priors learned by RF2NA, and preserve geometric symmetries during generation.

### 4.1 RNA SEQUENCE AND STRUCTURE REPRESENTATION

We represent RNA molecules through both discrete sequence and continuous structural components. The sequence $\mathbf{s} \in \{A, C, G, U, N\}^L$ consists of $L$ nucleotides from the five-letter RNA alphabet. For computational processing, we encode sequences as one-hot vectors $\mathbf{s}_{\text{oh}} \in \{0,1\}^{L \times 5}$ or categorical indices $\mathbf{s}_{\text{idx}} \in \{27, 28, 29, 30, 31\}^L$, for compatibility with RF2NA's encoding scheme. The 3D structure is represented through atomic Cartesian coordinates $\mathbf{X} \in \mathbb{R}^{L \times N_a \times 3}$, where $N_a$ denotes the number of atoms per nucleotide. Following crystallographic conventions, we track all heavy atoms in the RNA backbone including the phosphate group (P, OP1, OP2, O5') and sugar ribose (C5', C4', O4', C3', O3', C2', O2', C1'), as well as those in the base (purine: N9, C8, N7, C5, C6, N6/O6, N1,

C2, N3, C4; pyrimidine: N1, C2, O2, N3, C4, N4/O4, C5, C6). This yields up to $N_a = 27$ atom positions per nucleotide, with an associated binary mask $\mathbf{M} \in \{0, 1\}^{L \times N_a}$ indicating atom presence.

For each nucleotide $i$, we define a local coordinate frame $\mathcal{F}_i = (\mathbf{R}_i, \mathbf{t}_i)$ where $\mathbf{R}_i \in SO(3)$ represents the rotation matrix and $\mathbf{t}_i \in \mathbb{R}^3$ represents the translation vector. Following recent RNA parameterization (Morehead et al., 2023; Anand et al., 2025), these frames are constructed from three atoms per nucleotide (C4', C1', and the glycosidic nitrogen N1/N9) using the Gram-Schmidt process:

$$
\begin{aligned}
\mathbf{v}_1 &= \mathbf{x}_{C1'} - \mathbf{x}_{C4'}, \quad \mathbf{e}_1 = \mathbf{v}_1/\|\mathbf{v}_1\| \\
\mathbf{v}_2 &= \mathbf{x}_{N1/N9} - \mathbf{x}_{C4'}, \quad \mathbf{u}_2 = \mathbf{v}_2 - (\mathbf{v}_2 \cdot \mathbf{e}_1)\mathbf{e}_1 \\
\mathbf{e}_2 &= \mathbf{u}_2/\|\mathbf{u}_2\|, \quad \mathbf{e}_3 = \mathbf{e}_1 \times \mathbf{e}_2 \\
\mathbf{R}_i &= [\mathbf{e}_1, \mathbf{e}_2, \mathbf{e}_3], \quad \mathbf{t}_i = \mathbf{x}_{C4'}
\end{aligned}
\tag{2}
$$

The transformation between frames follows $\mathbf{x}_j^{(i)} = \mathbf{R}_i^T(\mathbf{x}_j - \mathbf{t}_i)$, where $\mathbf{x}_j^{(i)}$ represents position $j$ in frame $i$. This frame construction ensures $SE(3)$ equivariance, i.e., transformations of the global coordinates induce corresponding transformations of the local frames.

## 4.2 RoseTTAFold2NA as Pretrained Denoiser

RoseTTAFold2NA (RF2NA) (Baek et al., 2022) employs a three-track neural network architecture that simultaneously processes and updates three complementary representations of RNA-protein complexes. This architecture enables information flow between sequence, pairwise, and structural representations through iterative refinement blocks. The sequence track processes 1D features $\mathbf{h}^{(1D)} \in \mathbb{R}^{L \times d_{\text{seq}}}$ capturing positional and evolutionary information. The pair track maintains pairwise representations $\mathbf{h}^{(2D)} \in \mathbb{R}^{L \times L \times d_{\text{pair}}}$ encoding inter-residue relationships. The **structure track** operates on $SE(3)$-equivariant features $\mathbf{h}^{(3D)} \in \mathbb{R}^{L \times d_{\text{struct}}}$ coupled with coordinate frames $\{\mathcal{F}_i\}_{i=1}^L$. Information exchange between tracks occurs through attention-based communication modules. After $N_{\text{blocks}}$ refinement iterations, the model predicts per-residue frames and torsion angles. The final predictions include $\hat{\mathcal{F}}_i = (\hat{\mathbf{R}}_i, \hat{\mathbf{t}}_i)$, $\hat{\boldsymbol{\alpha}}_i = \{\phi_i, \psi_i, \chi_{i,1}, \ldots, \chi_{i,k}\}$, from which all-atom coordinates are reconstructed through geometric operations using idealized bond lengths and angles.

We reuse the pretrained RF2NA trunk as the shared representation and add diffusion heads including sequence head that outputs categorical logits, translation head that outputs per-residue translational noise, and rotation head that outputs per-nucleotide tangent velocities on $SO(3)$. Time-step embeddings $e(t)$ are injected into all tracks. We fine-tune these heads as well as the RF2NA model, initialized with pretrained weights, preserving the rich cross-molecular priors while improving sample-efficiency in the low-data RNA regime.

## 4.3 Joint Diffusion Framework

We extend RoseTTAFold2NA into a generative model by embedding it within a diffusion framework that jointly models discrete sequences and continuous structures. This requires specific treatment of the different data modalities, maintaining the coupling between sequence and structure throughout the diffusion process while respecting their distinct mathematical properties.

### 4.3.1 Discrete Sequence Diffusion

For RNA sequences, we implement a discrete diffusion (Sahoo et al., 2024; Shi et al., 2024) process that operates directly on categorical distributions. Following the absorbing state diffusion framework, we define transition matrices that progressively corrupt sequences toward a uniform distribution over nucleotides. The forward transition at timestep $t$ is parameterized by a matrix $\mathbf{Q}_t \in \mathbb{R}^{5 \times 5}$, $\mathbf{Q}_t = (1 - \beta_t^{\text{seq}})\mathbf{I} + \beta_t^{\text{seq}}\mathbf{U}$, where $\mathbf{I}$ is the identity matrix, $\mathbf{U}_{ij} = 1/5$ is the uniform transition matrix, and $\beta_t^{\text{seq}} \in [0, 1]$ controls the corruption rate. We employ a cosine schedule for $\beta_t^{\text{seq}}$, $\beta_t^{\text{seq}} = \cos\left(\frac{\pi}{2} \cdot \frac{t}{T}\right)^2$. The forward process defines the conditional distribution, $q(\mathbf{s}_t|\mathbf{s}_{t-1}) = \text{Categorical}(\mathbf{s}_t; \mathbf{Q}_t\mathbf{s}_{t-1})$. The marginal distribution at timestep $t$ can be computed in closed form

$$
q(\mathbf{s}_t|\mathbf{s}_0) = \text{Categorical}(\mathbf{s}_t; \bar{\mathbf{Q}}_t\mathbf{s}_0), \quad \bar{\mathbf{Q}}_t = \prod_{i=1}^t \mathbf{Q}_i.
\tag{3}
$$

The reverse process learns to predict clean sequence from noised version, $p_\theta(\mathbf{s}_{t-1}|\mathbf{s}_t, \mathbf{X}_t) = $ Categorical$(\mathbf{s}_{t-1}; \hat{\mathbf{s}}_0(\mathbf{s}_t, \mathbf{X}_t, t))$, where $\hat{\mathbf{s}}_0$ is model's prediction of clean sequence given the current noised state.

### 4.3.2 SE(3)-EQUIVARIANT STRUCTURE DIFFUSION

For three-dimensional structures, we implement $SE(3)$-equivariant diffusion that respects the symmetries of molecular systems. The goal is to define a diffusion process on the manifold of 3D structures that maintains equivariance under rotations and translations. We decompose each atomic position into a frame component and a position within that frame $\mathbf{x}_{i,a} = \mathbf{R}_i \mathbf{r}_{i,a} + \mathbf{t}_i$, where $\mathbf{r}_{i,a}$ represents the position of atom $a$ in the local frame of nucleotide $i$. The forward diffusion process operates separately on the frame orientations, frame translations, and local atomic positions (Watson et al., 2022). For frame translations $\mathbf{t} \in \mathbb{R}^3$, we apply standard Gaussian diffusion

$$q(\mathbf{t}_t|\mathbf{t}_0) = \mathcal{N}(\mathbf{t}_t; \sqrt{\bar{\alpha}_t^{\text{trans}}}\mathbf{t}_0, (1 - \bar{\alpha}_t^{\text{trans}})\mathbf{I}_3). \tag{4}$$

For frame rotations $\mathbf{R} \in SO(3)$, we employ the Isotropic Gaussian distribution on $SO(3)$ ($\mathcal{IG}_{SO(3)}$), which provides a natural diffusion process on the rotation manifold. The forward process follows

$$q(\mathbf{R}_t|\mathbf{R}_0) = \mathcal{IG}_{SO(3)}(\mathbf{R}_t; \mathbf{R}_0, \kappa_t), \tag{5}$$

where the concentration parameter $\kappa_t$ decreases with time, converging to a uniform distribution over $SO(3)$ as $t \to T$. The $\mathcal{IG}_{SO(3)}$ distribution has density $p(\mathbf{R}; \mathbf{R}_0, \kappa) \propto \exp(\kappa \cdot \text{tr}(\mathbf{R}_0^T \mathbf{R}))$.

To sample from this distribution, we use the axis-angle representation. Given a reference rotation $\mathbf{R}_0$, we sample a perturbation through the exponential map $\mathbf{R}_t = \mathbf{R}_0 \cdot \exp([\boldsymbol{\omega}]_\times)$, where $\boldsymbol{\omega} \sim \mathcal{N}(0, \sigma_t^2 \mathbf{I}_3)$ and $[\boldsymbol{\omega}]_\times$ denotes the skew-symmetric matrix formed from $\boldsymbol{\omega}$. The matrix exponential $\exp([\boldsymbol{\omega}]_\times)$ can be computed using Rodrigues' formula $\exp([\boldsymbol{\omega}]_\times) = \mathbf{I} + \frac{\sin\|\boldsymbol{\omega}\|}{\|\boldsymbol{\omega}\|}[\boldsymbol{\omega}]_\times + \frac{1-\cos\|\boldsymbol{\omega}\|}{\|\boldsymbol{\omega}\|^2}[\boldsymbol{\omega}]_\times^2$. The noise schedule for rotations follows $\sigma_t^2 = -2\log\bar{\alpha}_t^{\text{rot}}$, $\bar{\alpha}_t^{\text{rot}} = \prod_{s=1}^{t}\alpha_s^{\text{rot}}$. For local atomic positions within frames, we apply standard Gaussian noise $q(\mathbf{r}_{t,i,a}|\mathbf{r}_{0,i,a}) = \mathcal{N}(\mathbf{r}_{t,i,a}; \sqrt{\bar{\alpha}_t^{\text{local}}}\mathbf{r}_{0,i,a}, (1 - \bar{\alpha}_t^{\text{local}})\mathbf{I}_3)$. Thus the complete forward process for structures combines these components

$$q(\mathbf{X}_t|\mathbf{X}_0) = \prod_{i=1}^{L} q(\mathbf{R}_{t,i}|\mathbf{R}_{0,i}) \cdot q(\mathbf{t}_{t,i}|\mathbf{t}_{0,i}) \cdot \prod_{a=1}^{N_a} q(\mathbf{r}_{t,i,a}|\mathbf{r}_{0,i,a}). \tag{6}$$

### 4.3.3 JOINT REVERSE PROCESS

The reverse process $p_\theta(\mathbf{s}_{t-1}, \mathbf{X}_{t-1}|\mathbf{s}_t, \mathbf{X}_t)$ is parameterized by RF2NA to jointly denoise sequences and structures. Given noised inputs $(\mathbf{s}_t, \mathbf{X}_t)$ at timestep $t$, the model predicts the clean data $\hat{\mathbf{s}}_0, \hat{\mathbf{X}}_0 = f_{\text{RF2NA}}(\mathbf{s}_t, \mathbf{X}_t, t)$. The crucial property is that this prediction maintains $SE(3)$-**equivariance**. For any transformation $g = (\mathbf{R}_g, \mathbf{t}_g) \in SE(3)$, $f_{\text{RF2NA}}(\mathbf{s}_t, g \cdot \mathbf{X}_t, t) = (\hat{\mathbf{s}}_0, g \cdot \hat{\mathbf{X}}_0)$. This equivariance is guaranteed by the architecture of RF2NA, which processes geometric information only through invariant features (distances, angles) and equivariant operations (frame transformations). To sample the reverse transitions, we use the predicted clean data to parameterize the distributions. For sequences

$$p_\theta(\mathbf{s}_{t-1}|\mathbf{s}_t, \mathbf{X}_t) = \sum_{\mathbf{s}_0} q(\mathbf{s}_{t-1}|\mathbf{s}_t, \mathbf{s}_0) \cdot p_\theta(\mathbf{s}_0|\mathbf{s}_t, \mathbf{X}_t), \tag{7}$$

where $q(\mathbf{s}_{t-1}|\mathbf{s}_t, \mathbf{s}_0)$ is the posterior transition probability that can be computed using Bayes' rule. For structures, we compute the reverse transition using the score function

$$p_\theta(\mathbf{X}_{t-1}|\mathbf{X}_t, \mathbf{s}_t) = \mathcal{N}(\mathbf{X}_{t-1}; \boldsymbol{\mu}_\theta(\mathbf{X}_t, \mathbf{s}_t, t), \boldsymbol{\Sigma}_t), \tag{8}$$

where the mean is computed using the predicted clean structure $\boldsymbol{\mu}_\theta = \frac{\sqrt{\bar{\alpha}_{t-1}}\beta_t}{1-\bar{\alpha}_t}\hat{\mathbf{X}}_0 + \frac{\sqrt{\alpha_t}(1-\bar{\alpha}_{t-1})}{1-\bar{\alpha}_t}\mathbf{X}_t$.

### 4.4 TRAINING OBJECTIVE

We train the model to reconstruct clean data from noised inputs through a combined loss function that balances sequence accuracy, structural precision, and physical validity. The total objective combines

multiple terms to ensure both local and global consistency

$$\mathcal{L}_{\text{total}} = \lambda_{\text{seq}} * \mathcal{L}_{\text{seq}} + \lambda_{\text{str}} * \mathcal{L}_{\text{str}} + \lambda_{\text{rmsd}} * \mathcal{L}_{\text{rmsd}} + \lambda_{\text{geom}} * \mathcal{L}_{\text{geom}} + \lambda_{\text{lj}} * \mathcal{L}_{\text{lj}} \tag{9}$$

$$= -\lambda_{\text{seq}} * \sum_{i=1}^{L} \log p_\theta(\mathbf{s}_{0,i}|\mathbf{s}_{t,i}, \mathbf{X}_t, t) + \frac{\lambda_{\text{str}}}{L^2} \sum_{i,j} |\mathcal{F}_i^{-1}(\hat{\mathbf{x}}_j) - \mathcal{F}_i^{-1}(\mathbf{x}_j)|_2 + \lambda_{\text{lj}} * \mathcal{L}_{\text{lj}} \tag{10}$$

$$+ \lambda_{\text{rmsd}} * \sqrt{\frac{1}{N} \sum_{a=1}^{N_{\text{atoms}}} |\hat{\mathbf{x}}_a - \mathbf{x}_a|^2} + \lambda_{\text{geom}} * \left( \sum_{\text{bonds}} (\|\hat{\mathbf{b}}\| - b_0)^2 + \sum_{\text{angles}} (\cos\hat{\theta} - \cos\theta_0)^2 \right). \tag{11}$$

The sequence loss employs cross-entropy over nucleotide predictions, where $\mathbf{s}_{0,i}$ is the true nucleotide at position $i$; the structure loss measures frame-aligned point error (FAPE), ensuring accurate local geometry, where $\mathcal{F}_i^{-1}$ transforms coordinates into the local frame of nucleotide $i$; the coordinate RMSD loss enforces global structural accuracy after optimal superposition, and the geometry loss maintains proper local stereochemistry by bond lengths and angles, where $b_0$ and $\theta_0$ represent ground truth values from structural databases. The Lennard-Jones loss prevents steric clashes while maintaining appropriate van der Waals interactions

$$\mathcal{L}_{\text{lj}} = \sum_{i<j} \begin{cases} \epsilon \left[ \left( \frac{\sigma_{ij}}{r_{ij}} \right)^{12} - 2 \left( \frac{\sigma_{ij}}{r_{ij}} \right)^{6} \right] & r_{ij} < r_{\text{cutoff}} \\ 0 & \text{otherwise} \end{cases} \tag{12}$$

where $r_{ij} = \|\hat{\mathbf{x}}_i - \hat{\mathbf{x}}_j\|$, and $\sigma_{ij}$ represents the sum of van der Waals radii.

During training, we employ an iterative refinement strategy where the model learns to progressively denoise from various noise levels. We sample timesteps $t \sim \mathcal{U}(1, T)$ and apply stochastic masking to enable flexible conditioning $\mathbf{m}_{\text{seq}} \sim \text{Bernoulli}(p_{\text{mask}})$, $\mathbf{m}_{\text{str}} \sim \text{Bernoulli}(p_{\text{mask}})$. This allows the model to learn multiple generation modes: sequence-to-structure prediction ($\mathbf{m}_{\text{seq}} = \mathbf{0}, \mathbf{m}_{\text{str}} = \mathbf{1}$), inverse folding ($\mathbf{m}_{\text{seq}} = \mathbf{1}, \mathbf{m}_{\text{str}} = \mathbf{0}$), and joint generation ($\mathbf{m}_{\text{seq}} = \mathbf{m}_{\text{str}} = \mathbf{1}$). We study the influence on model learning of different sequence and structure masking strategy in Appendix H.

## 4.5 CONDITIONAL GENERATION FOR RNA-PROTEIN COMPLEX DESIGN

Beyond joint sequence-structure co-generation for isolated RNA molecules, our framework naturally extends to conditional generation tasks where RNA molecules are designed to interact with specific protein targets. This capability addresses critical applications in therapeutic design, where RNA aptamers, riboswitches, or regulatory elements must be engineered to bind predetermined protein partners with high affinity and specificity.

In the conditional generation setting, we partition the full molecular system into protein and RNA components $\mathcal{M} = \{\mathcal{P}, \mathcal{R}\}$, where $\mathcal{P}$ represents the fixed protein target with structure $\mathbf{X}^{\text{prot}} \in \mathbb{R}^{L_p \times N_a^p \times 3}$ and sequence $\mathbf{s}^{\text{prot}} \in \mathcal{A}_{\text{protein}}^{L_p}$, while $\mathcal{R}$ denotes the RNA to be designed with length $L_r$. The conditional diffusion process modifies the standard joint generation to respect the protein constraint. During the forward process, we apply noise only to the RNA components while preserving the protein structure

$$q(\mathbf{X}_t^{\text{RNA}}, \mathbf{s}_t^{\text{RNA}}|\mathbf{X}_0^{\text{RNA}}, \mathbf{s}_0^{\text{RNA}}, \mathbf{X}^{\text{prot}}, \mathbf{s}^{\text{prot}}) = q(\mathbf{X}_t^{\text{RNA}}|\mathbf{X}_0^{\text{RNA}}) \cdot q(\mathbf{s}_t^{\text{RNA}}|\mathbf{s}_0^{\text{RNA}}). \tag{13}$$

The reverse process learns to denoise the RNA components conditioned on the protein context $p_\theta(\mathbf{X}_{t-1}^{\text{RNA}}, \mathbf{s}_{t-1}^{\text{RNA}}|\mathbf{X}_t^{\text{RNA}}, \mathbf{s}_t^{\text{RNA}}, \mathbf{X}^{\text{prot}}, \mathbf{s}^{\text{prot}}, t)$. Crucially, the three-track architecture of RoseTTAFold2NA naturally accommodates this conditioning through its pair representation, which encodes inter-molecular interactions. The pair features between protein and RNA residues capture potential binding interfaces $\mathbf{h}_{ij}^{\text{inter}} = f_{\text{bind}}(\mathbf{h}_i^{\text{prot}}, \mathbf{h}_j^{\text{RNA}}) + f_{\text{geom}}(\mathbf{X}_i^{\text{prot}}, \mathbf{X}_j^{\text{RNA}})$. This conditional generation framework enables the design of RNA molecules with tailored properties, opening avenues for computational design of RNA-based therapeutics, biosensors, and synthetic regulatory circuits that interface with specific protein targets and beyond.

## 4.6 RL-ENHANCED DIFFUSION INFERENCE

To improve generation quality at inference time, we introduce value-based importance sampling (SVDD (Li et al., 2024b)), which leverages the model's learned representations to guide the reverse

diffusion process toward high-quality samples (Uehara et al., 2025; Li et al., 2025; Su et al., 2025b). This approach draws inspiration from reinforcement learning, treating the generation process as a sequential decision problem where each denoising step can be optimized based on expected downstream performance. Given a partially denoised sample $(\mathbf{X}_t, \mathbf{s}_t)$ at timestep $t$, we generate $M$ candidate next states through the standard reverse process

$$(\mathbf{X}_{t-1}^{(m)}, \mathbf{s}_{t-1}^{(m)}) \sim p_\theta(\cdot | \mathbf{X}_t, \mathbf{s}_t, t), \quad m = 1, \ldots, M. \tag{14}$$

For each candidate, we compute a reward signal $r_m$ that evaluates sample quality. We select the best candidate based on the reward task,

$$m^* = \arg\max_m \left[ r^{(m)}(\mathbf{X}_0^{(m)}, \mathbf{s}_0^{(m)} | \mathbf{X}_{t-1}^{(m)}, \mathbf{s}_{t-1}^{(m)}) + \tau \log p_\theta(\mathbf{X}_{t-1}^{(m)}, \mathbf{s}_{t-1}^{(m)} | \mathbf{X}_t, \mathbf{s}_t, t) \right], \tag{15}$$

where $\tau$ controls the trade-off between reward optimization and staying close to the learned distribution. We employ multiple reward functions depending on the design objective, as further explained in Appendix E.2. At inference time, the conditional generation process can also be further enhanced through interface-focused reward. This enhanced inference procedure significantly improves the quality and diversity of generated RNA designs while maintaining computational tractability through selective application of the verification steps.

## 5 EXPERIMENTAL STUDIES

We evaluate the proposed joint sequence-structure diffusion framework on three task settings, (1) single RNA co-design, (2) RNA-protein complex co-design, and (3) protein-conditioned RNA binder co-design. We explain our settings and protocol, followed by experimental analyses on each task, while more experimental details and results are provided in Appendix F and G.

### 5.1 SETTINGS

**Dataset and split.** We adopt two datasets for our experiments, the RNASolo dataset and the protein-RNA complex dataset. For the single RNA task, we leverage the RNASolo (Adamczyk et al., 2022) dataset which consists of extracted individual RNA 3D structures from the Protein Databank (PDB) (Berman et al., 2000). We filter the full dataset to resolution $< 4\text{Å}$, resulting in a total dataset size of 15k data points. We cluster the RNA sequences into groups based on structural similarity using US-align (Zhang et al., 2022) with a similarity threshold of TM-score $> 0.45$ for clustering, and split the train/validation and test sets on structurally dissimilar clusters in 40:1:1 ratio, following Joshi et al. (2025). For the complex and conditional tasks, protein-RNA complexes from the PDBBind dataset (2020 version) are used for training and evaluation following Liu et al. (2017); Nori & Jin (2024). Experiments are performed on two splits. The first RF2NA pre-training split assigns the RF2NA validation and test sets as the test split while remaining samples are randomly split into training and validation with 9:1 ratio, and the second RNA sequence similarity split clusters RNA chains using CD-HIT (Fu et al., 2012; Joshi et al., 2025) and splits clusters randomly into train, validation, and test in 8:1:1 ratio. Further dataset details are included in the Appendix.

**Metrics.** We assess model performance using multiple structural- and sequence-based metrics that capture both fidelity and self-consistency following Morehead et al. (2023). To assess self-consistent designability, we report structural similarities in terms of self-consistency RMSD (scRMSD) and TM-score (scTM), which measures consistency between generated sequence-structure pairs by comparing the co-designed structure with the structure predicted from the generated sequence. We also report Local Distance Difference Test (lDDT), which measures how well inter-atomic distances in the co-designed structure match those in the sequence-predicted reference structure. Self-consistency sequence recovery rate (scSeqRec) measures the percentage of recovered nucleotides between a co-designed sequence and the inverse-folded sequence given its co-designed 3D structure backbone. For diversity, we report qTMclust diversity, which measures the fraction of distinct structural clusters identified under structural alignment (Zhang et al., 2022) (TM-cutoff $0.45$). For single RNA task, we further report the structural success rate, with scRMSD $< 5\text{Å}$ denoting a successfully-designed macromolecule. For conditional tasks, following Nori & Jin (2024), we report ground truth RMSD (GT-RMSD) and sequence recovery rate (GT-SeqRec) and lDDT measured against the reference RNA structure and sequence in the complex of the condition. Note that RMSD is calculated for

Table 1: Comparison across methods on single RNA task. Success rate is the percentage of samples with scRMSD $< 5$Å. qTMclust diversity uses TM-cutoff 0.45. Average value and standard deviation are reported for all metrics. For MMDIFF, we rerun its official implementation under our setting.

| Method | Success rate (% ↑) | scRMSD (Å ↓) | scTM-score (↑) | LDDT (↑) | scSeqRec (% ↑) | qTMclust Diversity (↑) |
|---|---|---|---|---|---|---|
| Random generation | 0.00±0.00 | 39.74±4.82 | 0.05±0.03 | 0.23±0.05 | 1.06±0.40 | 0.99±0.01 |
| MMDIFF | 8.86±3.12 | 35.77±5.15 | 0.12±0.06 | 0.33±0.07 | 23.90±8.42 | **1.00**±0.00 |
| RNA-FrameFlow + gRNAde | 15.52±4.33 | 18.65±4.27 | 0.32±0.08 | 0.43±0.12 | 45.65±2.23 | 0.76±0.10 |
| RiboDiff | **97.38**±4.86 | **3.43**±0.51 | **0.71**±0.04 | **0.74**±0.06 | **48.57**±4.20 | **1.00**±0.00 |

Table 2: Comparison across methods on RNA-protein complex. Average value and standard deviation are reported for all metrics.

| Method | scRMSD (Å ↓) | scTM-score (↑) | LDDT (↑) | scSeqRec (% ↑) | qTMclust Diversity (↑) |
|---|---|---|---|---|---|
| Random generation | 43.51±6.26 | 0.002±0.003 | 0.26±0.06 | 0.59±0.83 | **1.00**±0.00 |
| MMDIFF | 30.84±4.93 | 0.015±0.010 | 0.38±0.07 | 17.46±6.15 | 0.96±0.02 |
| RiboDiff | **7.43**±0.88 | **0.422**±0.073 | **0.71**±0.05 | **52.91**±3.90 | **1.00**±0.00 |

all atoms instead of backbone atoms. We also partially report interface confidence score ipTM, using Chai-Lab (Chai-1) (team et al., 2024) to evaluate binding affinity for designed RNA-protein complexes. Self-consistency, diversity and more interface metrics are reported in Appendix G.

**Baselines and protocol.** For single-RNA and complex co-design, we compare[1] with MMDiff, adapting its official implementation under our settings, and a random generation baseline, implemented by random model initialization. We also include a baseline that first designs backbones with RNA-FrameFlow and then applies gRNAde to generate sequences for those backbones, following our evaluation protocol. For protein-conditioned design, we compare to RNAFlow (Nori & Jin, 2024)[2] and a conditional variant of MMDiff. Following prior work, we sample 10 RNA designs per complex in the test set, conditioning on the protein backbone and sequence.

## 5.2 RESULTS

**Single RNA co-design.** Table 1 summarizes results on RNASolo. Our method achieves a success rate of 97.38%. Average scRMSD drops from 35.7 Å (MMDiff) to 3.43 Å with our model, a $> \mathbf{10\times}$ reduction, while scTM-score rises $\mathbf{6\times}$ higher and lDDT more than doubles. The improvement across all self-consistency metrics supports the central premise of our approach that co-optimizing sequence and structure with a pretrained denoiser produces structures that are simultaneously geometrically accurate and sequence-realizable. Whereas training-from-scratch diffusion struggles in the low-data RNA regime, reusing RF2NA's cross-molecular priors provides substantial inductive bias. Self-consistency in sequence also improves significantly, indicating that designed backbones admit sequences re-folding into highly similar structures. Diversity remains stably high, showing that our large quality gains are not obtained by mode-collapse toward a small set of folds. Figure 1 show sample visualizations, with more discussed in Appendix I.

**RNA-protein complex co-design.** On RNA-protein complexes, Table 2 shows large improvements in self-consistency over MMDiff, highlighting that our jointly generated RNAs are more design-consistent in complex environments. These gains reflect two coupled advantages of our formulation. First, $SE(3)$-equivariant diffusion respects global symmetries and improves interface geometry. Second, RF2NA tracks encode inter-chain couplings learned from protein-nucleic acid complexes, thus conditioning on the protein context shapes the RNA's denoising trajectory toward interface-compatible conformations.

**Protein-conditioned RNA binder co-design.** We next condition on a fixed protein and co-design an RNA binder. Table 3 reports ground-truth comparisons on the RF2NA pre-training split and the sequence-similarity split. Our method approximately doubles the sequence recovery, attains better GT-RMSD, and markedly raises LDDT, outperforming RNAFlow and conditional MMDiff across all metrics. Observations emerge that absolute GT-RMSD remains in the low tens of angstroms

---

[1]For another existing work RiboGen, due to lack of implementation detail and numerical results, we perform visual comparison in Appendix G.2

[2]We run RNAFlow-Base, which is the only implementation presented in the official code.

Table 3: Comparison across methods on conditional RNA co-design. For MMDiff and RNAFlow, we rerun their official implementation. Note that RMSD is calculated for all atoms.

| Method | RF2NA Pre-Training Split | | | Sequence Similarity Split | | |
| --- | --- | --- | --- | --- | --- | --- |
| | GT-SeqRec (% ↑) | GT-RMSD (Å↓) | LDDT (↑) | GT-SeqRec (% ↑) | GT-RMSD (Å↓) | LDDT (↑) |
| Conditional MMDiff | 24 ± 2 | 16.17 ± 2.15 | 0.35 ± 0.03 | 22 ± 2 | 17.91 ± 0.93 | 0.36 ± 0.02 |
| RNAFlow-Base | 27.98 ± 2.96 | 15.39 ± 1.67 | 0.40 ± 0.01 | 29.37 ± 1.23 | 17.48 ± 1.10 | 0.46 ± 0.01 |
| **RiboDiff** | **56.26 ± 1.58** | **13.20 ± 0.17** | **0.73 ± 0.02** | **53.54 ± 3.71** | **13.39 ± 0.22** | **0.71 ± 0.04** |

Table 4: Comparison across rewards with and w/o RL-based inference enhancement. We report the average and standard deviation results under $M = 10$.

| Reward | scRMSD (Å ↓) - Complex | GT-RMSD (Å ↓) - Conditional | ipTM (↑)- Conditional |
| --- | --- | --- | --- |
| RiboDiff | 7.43±0.88 | 13.20±0.17 | 0.166 ± 0.056 |
| RiboDiff+SVDD | 6.41±1.56 | 12.43±0.68 | 0.187 ± 0.084 |

across all methods, reflecting the inherent difficulty of recovering exact bound RNA conformations given structure flexibility. Nonetheless, our method consistently yields higher local confidence and much higher sequence agreement, suggesting that the generated binders are relatively well-formed. Performance remains stable under the sequence-similarity split, indicating generalization beyond close homologs. This robustness is consistent with leveraging a pretrained denoiser rather than training a generator from scratch on a small RNA-protein corpus.

**Effect of RL-style inference enhancement.** We assess the effect of inference enhancement SVDD (Sec. 4.6) on complexes and conditional binders. Table 4 shows consistent improvements without any parameter updates, indicating that reward-guided selection can effectively steer samples toward better interfaces on top of the learned diffusion prior. We use modest proposal counts ($M$) to keep runtime practical, while more proposals can further improve the properties with diminishing returns.

## 6    CONCLUSION AND DISCUSSION

We introduced a unified pretraining-guided diffusion framework for RNA sequence-structure co-design using RoseTTAFold2NA as the denoiser to inject rich cross-molecular priors. This joint formulation enables the model to synchronize sequence constraints with tertiary geometry throughout sampling, supports flexible conditioning, and benefits from RL-style inference guidance to further improve properties without retraining. Empirically, the approach achieves success on single-RNA co-design, large gains in compatibility for RNA-protein complexes, and strong ground-truth agreement on conditional binding, all while preserving sequence and structural diversity. Future directions include tighter integration of learned and physics-based energies (e.g., differentiable solvation/electrostatics), schedule-free or flow-diffusion hybrids for faster sampling, uncertainty calibration and active learning with wet-lab feedback, and broader conditioning (small molecules, multi-chain RNAs, or dynamic conformational ensembles). Overall, our results establish pretraining-guided joint diffusion as a powerful paradigm for programmable RNA design under data scarcity.

ACKNOWLEDGMENTS

This work was supported in part by National Institutes of Health under grant U01AG070112 and Advanced Research Projects Agency for Health under 1AY1AX000053.

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

## A   BROADER IMPACT

Generative co-design of RNAs has immediate relevance for therapeutics, synthetic biology, and diagnostics. By emphasizing symmetry, joint modeling, and pretrained priors, this work points to a general recipe for biomolecular design under data scarcity. We adhered to structure- and sequence-level OOD splits, reported both fidelity and diversity, and ablated inference-time guidance. We anticipate that open benchmarks coupling sequence, structure, and function will be essential for the community to converge on standardized protocols and to translate computational gains into experimentally validated advances. While this research primarily contributes to technical advancements in generative modeling, it has potential implications in domains such as therapeutics and biomolecular engineering. We acknowledge that generative models, particularly those optimized for specific biological domain, could be misused if not carefully applied. However, we emphasize the importance of responsible deployment and alignment with ethical guidelines in generative AI. Overall, our contributions align with the broader goal of machine learning methodologies, and we do not foresee any immediate ethical concerns beyond those generally associated with generative AI.

## B   LIMITATIONS AND FUTURE WORK

Despite strong relative gains, several limitations remain. Ground-truth RMSD for conditional binders is still improvable, reflecting both docking flexibility and the scarcity of resolved interfaces. Small absolute errors at the interface can dominate functional outcomes, and the flexibility of RNA structures calls for more accurate metrics, such as ipTM. Very long RNAs remain challenging due to accumulated torsion noise and long-range coaxial stacking, where we observed occasional failures to recover global register. In addition, our evaluation relies in part on self-consistency via structure prediction; although stringent and informative, this can induce alignment with predictor-specific biases. Also, the current model does not explicitly treat ions, cofactors, or solvent, which are known to stabilize key tertiary contacts in RNA; implicit geometry and clash penalties only partially capture these effects.

Models including RF2NA already move toward a unified structural prior over proteins, RNA, DNA, ligands and complexes. It is natural to expect analogous unified generative models that can design multi-component systems (e.g., protein-RNA-DNA complexes) within a single framework. This can be a large language model (LLM), which has shown great scientific capabilities recently (OpenAI, 2025; Yang et al., 2025; Huang et al., 2024b). A fully unified all-atom generative model would need to simultaneously cover very diverse chemistries and interfaces (proteins, nucleic acids, small molecules) (Li et al., 2024a; Fu et al., 2024; Yan et al., 2024; Su et al., 2025a), potentially diluting capacity for RNA if not carefully architected. Conceptually, unified generative models are highly desirable for designing multi-modal assemblies (e.g., ribonucleoprotein machines, CRISPR complexes, RNA-ligand systems). In this work, we focus on RNA and RNA-protein co-design under data scarcity. RF2NA provides rich cross-molecular priors tailored to these domains, and our diffusion layers are explicitly designed to respect RNA's discrete alphabet and SE(3) symmetries. Our work can be viewed as a step in this direction, since we already leverage a unified biomolecular predictor and show that pretraining-guided diffusion is an effective recipe under data scarcity. A natural extension is to generalize RiboDiff's dual discrete-continuous diffusion to multiple residue types and chains, effectively moving toward the unified design framework envisioned in the question.

## C   LLM USAGE STATEMENT

This manuscript used ChatGPT (GPT-5) for grammar correction, sentence structure improvement, and language clarity enhancement. All ideas, data interpretation, and scientific contributions remain solely the work of the authors.

## D   DIFFERENCES WITH CLOSE RELATED WORKS

RFdiffusion pioneered the "pretrained structure predictor to diffusion generator" paradigm for proteins. Our work is inspired by this paradigm, and we further extends to the RNA domains, which is substantially more data-limited and structurally distinct, and develops a joint sequence-all-atom structure

diffusion framework. We would like to highlight several key differences. Joint discrete–continuous co-diffusion for RNA sequence + all-atom structure. RFdiffusion primarily generates continuous backbone coordinates. In contrast, RiboDiff jointly diffuses over discrete sequences and all-atom SE(3)-equivariant frames, with a coupled objective that enforces sequence–structure co-optimisation throughout the trajectory. This enables us to more directly optimize self-consistency metrics that require both a realizable sequence and a physically plausible structure. We extend the RF2NA architecture to conditional generation of RNA binders for fixed proteins, leveraging its three-track representation to encode protein-RNA interfaces. To our knowledge, this is the first pretraining-guided diffusion model that performs protein-conditioned RNA binder co-design, and we show strong improvements over RNAFlow and conditional MMDiff. We integrate SVDD (soft value-based decoding) as an RL-inspired inference scheme tailored to RNA sequence–structure design, with rewards based on self-consistency and interface confidence. RFdiffusion operates purely on proteins using RoseTTAFold, whereas RiboDiff embeds RoseTTAFold2NA, which is trained on proteins, RNA, DNA, and their complexes. We therefore exploit cross-molecular priors that RFdiffusion does not study. Also, RNA's conformational flexibility and sparse structural data make this extension experimentally non-trivial. Our results show that a pretraining-guided approach yields >10× lower scRMSD vs MMDiff on single RNA and large gains on complexes.

Our high-level conceptual template of joint discrete + continuous generative modeling is related to DFM-Multiflow (Campbell et al., 2024). Our framework differs from the DFM-Multiflow work in several important ways, both methodologically and in scope, beyond a straightforward adaptation to RNA. The DFM-Multiflow work learns a DFM for the sequence and a continuous-time flow model, FrameFlow, for protein structures, trained from scratch. Our work uses a dual diffusion process (discrete diffusion on nucleotides + SE(3)-equivariant diffusion on rigid frames) with RF2NA embedded as the pretrained denoiser prior. This yields a pre-trained multi-context trunk capable of handling proteins, RNAs, DNA, and complexes, and a diffusion parameterization that explicitly matches RF2NA's rigid-frame representation and internal three-track structure. This "pretrained predictor as denoiser" formulation and the way we adapt RF2NA's architecture for generative RNA (and complex/conditional) co-design is substantially different from DFM-Multiflow's from-scratch protein flow model. The DFM-Multiflow work mainly positions with the insight that discrete flow-based model can be realized using continuous time markov chains, and use one DFM and one FrameFlow model to form Multiflow, which is a different focus from our paper. We embeds RF2NA as the denoiser into a dual diffusion model with the insight that injecting rich cross-molecular priors while enabling sample-efficient learning from limited RNA data. DFM-Multiflow focuses on single protein co-design. In contrast, our framework exploits RF2NA's multi-context backbone to support three tasks in the framework, single RNA design, RNA-protein complex design, and Protein-conditioned RNA binder design. This unified, cross-molecular generative scope is, to our knowledge, not covered by GF-DSS. We incorporate RL-style guidance (value-based decoding) into the diffusion sampling process, using task-aligned non-differentiable rewards such as interface quality, and ipTM-like scores. DFM-Multiflow does not perform reward-guided inference on design objectives. RNA generative modeling presents challenges not handled in DFM-Multiflow. Our model jointly diffuses all-atom coordinates and discrete nucleotides, together with geometric and steric losses (e.g., Lennard-Jones) tailored to RNA. On the empirical side, we show that integrating RF2NA into a dual diffusion framework yields significant gains over joint RNA diffusion models trained from scratch and two-stage pipelines. This systematically demonstrates, for the first time, that pretraining-guided joint diffusion over RF2NA is a powerful and practical recipe for RNA design under data scarcity.

# E  METHODOLOGY DETAILS

## E.1  ROSETTAFOLD2NA AS PRETRAINED DENOISER

RoseTTAFold2NA (RF2NA) (Baek et al., 2022) employs a three-track neural network architecture that simultaneously processes and updates three complementary representations of RNA-protein complexes. This architecture enables information flow between sequence, pairwise, and structural representations through iterative refinement blocks.

The **sequence track** processes 1D features $\mathbf{h}^{(1D)} \in \mathbb{R}^{L \times d_{\text{seq}}}$ capturing positional and evolutionary information. Initial embeddings combine sequence encoding with positional features

$$\mathbf{h}_i^{(1D)} = f_{\text{embed}}(\mathbf{s}_i) + f_{\text{pos}}(i) + f_{\text{type}}(\tau_i), \tag{16}$$

where $\tau_i$ indicates residue type (RNA/protein). The **pair track** maintains pairwise representations $\mathbf{h}^{(2D)} \in \mathbb{R}^{L \times L \times d_{\text{pair}}}$ encoding inter-residue relationships. These features are initialized from multiple sequence alignments (MSAs) and relative positional encodings

$$\mathbf{h}_{ij}^{(2D)} = f_{\text{pair}}(\text{MSA}_{ij}) + f_{\text{rel}}(i - j) + f_{\text{dist}}(\|\mathbf{x}_i - \mathbf{x}_j\|). \tag{17}$$

The **structure track** operates on $SE(3)$-equivariant features $\mathbf{h}^{(3D)} \in \mathbb{R}^{L \times d_{\text{struct}}}$ coupled with coordinate frames $\{\mathcal{F}_i\}_{i=1}^L$. This track employs SE(3)-equivariant transformers that preserve rotational and translational symmetries

$$\mathbf{h}^{(3D)}, \{\mathcal{F}_i\} = \text{SE3-Transformer}(\mathbf{h}^{(3D)}, \{\mathcal{F}_i\}, \mathbf{E}), \tag{18}$$

where edge features $\mathbf{E}$ encode geometric relationships.

Information exchange between tracks occurs through attention-based communication modules. The sequence-to-pair update employs outer product mean

$$\Delta\mathbf{h}_{ij}^{(2D)} = \text{Linear}(\mathbf{h}_i^{(1D)} \otimes \mathbf{h}_j^{(1D)}). \tag{19}$$

The pair-to-structure update aggregates pairwise information

$$\Delta\mathbf{h}_i^{(3D)} = \sum_j \text{Attention}(\mathbf{h}_i^{(3D)}, \mathbf{h}_j^{(3D)}, \mathbf{h}_{ij}^{(2D)}). \tag{20}$$

After $N_{\text{blocks}}$ refinement iterations, the model predicts per-residue frames and torsion angles. The final predictions include

$$\hat{\mathcal{F}}_i = (\hat{\mathbf{R}}_i, \hat{\mathbf{t}}_i), \quad \hat{\boldsymbol{\alpha}}_i = \{\phi_i, \psi_i, \chi_{i,1}, \ldots, \chi_{i,k}\}, \tag{21}$$

from which all-atom coordinates are reconstructed through geometric operations using idealized bond lengths and angles.

We reuse the pretrained RF2NA trunk as the shared representation and add diffusion heads including sequence head that outputs categorical logits over $\{A, C, G, U, N\}$, translation head that outputs per-residue translational noise (or $x_0$-prediction), and rotation head that outputs per-nucleotide tangent velocities on $SO(3)$. Time-step embeddings $e(t)$ are injected into all tracks. We fine-tune these heads as well as the RF2NA model, initialized with pretrained weights, preserving the rich cross-molecular priors while improving sample-efficiency in the low-data RNA regime.

## E.2 RL-Enhanced Diffusion Inference

To improve generation quality at inference time, we introduce value-based importance sampling (SVDD), which leverages the model's learned representations to guide the reverse diffusion process toward high-quality samples. This approach draws inspiration from reinforcement learning, treating the generation process as a sequential decision problem where each denoising step can be optimized based on expected downstream performance. Given a partially denoised sample $(\mathbf{X}_t, \mathbf{s}_t)$ at timestep $t$, we generate $M$ candidate next states through the standard reverse process

$$(\mathbf{X}_{t-1}^{(m)}, \mathbf{s}_{t-1}^{(m)}) \sim p_\theta(\cdot|\mathbf{X}_t, \mathbf{s}_t, t), \quad m = 1, \ldots, M. \tag{22}$$

For each candidate, we compute a reward signal $r_m$ that evaluates sample quality. We select the best candidate based on the reward task,

$$m^* = \arg\max_m \left[ r^{(m)}(\mathbf{X}_0^{(m)}, \mathbf{s}_0^{(m)}|\mathbf{X}_{t-1}^{(m)}, \mathbf{s}_{t-1}^{(m)}) + \tau \log p_\theta(\mathbf{X}_{t-1}^{(m)}, \mathbf{s}_{t-1}^{(m)}|\mathbf{X}_t, \mathbf{s}_t, t) \right], \tag{23}$$

where $\tau$ controls the trade-off between reward optimization and staying close to the learned distribution.

We employ multiple reward functions depending on the design objective. The **self-consistency reward** measures agreement between the generated structure and the model's structure prediction from sequence alone

$$r_{\text{sc}}^{(m)} = -\text{scRMSD}(\mathbf{X}_{t-1}^{(m)}, f_{\text{fold}}(\mathbf{s}_{t-1}^{(m)})),  \tag{24}$$

where $f_{\text{fold}}$ represents the pre-trained folding model. The **predicted confidence reward** leverages the model's internal confidence metrics

$$r_{\text{conf}}^{(m)} = \text{pLDDT}(\mathbf{X}t-1^{(m)}, \mathbf{s}t-1^{(m)}) + \lambda_{\text{tm}}\text{pTM}(\mathbf{X}t-1^{(m)}, \mathbf{s}t-1^{(m)})  \tag{25}$$

The interface predicted TM-score (ipTM) evaluates the quality of predicted inter-molecular contacts

$$\text{ipTM} = \frac{1}{L_{\text{interface}}} \sum_{i \in \mathcal{P}, j \in \mathcal{R}} \frac{1}{1 + \left(\frac{d_{ij} - d_{ij}^{\text{pred}}}{d_0}\right)^2} \cdot w_{ij},  \tag{26}$$

where $d_{ij}$ represents inter-molecular distances, $d_0$ is a normalization factor, and $w_{ij}$ weights contacts by their predicted confidence. The **complex stability reward** (for RNA-protein complexes) evaluates binding interfaces

$$r_{\text{bind}}^{(m)} = \text{ipTM}(\mathbf{X}\text{RNA}^{(m)}, \mathbf{X}\text{protein}^{(m)}) - \beta \cdot E_{\text{clash}}(\mathbf{X}_{t-1}^{(m)})  \tag{27}$$

At inference time, the conditional generation process can also be further enhanced through interface-focused reward.

## F  EXPERIMENTAL DETAILS

### F.1  DATASET DETAILS

We curate the single RNA dataset using RNASolo (Adamczyk et al., 2022), a repository of RNA 3D structures extracted from solo RNAs, protein-RNA complexes, and DNA-RNA hybrids in the PDB. We used all RNA structures at resolution <4.0Å resulting in 4k+ unique RNA sequences for which a total of 15k+ structures are available (RNASolo date cutoff: November 2024). Our training data contains 3 Million unique nucleotides. Further dataset statistics are available in Figure 2, illustrating the diversity of our dataset in terms of sequence length. We cluster the unique RNAs into groups based on structural similarity. We use US-align with a similarity threshold of TM-score >0.45 for clustering, and ensure that we train and test on structurally dissimilar clusters. After clustering, we split the RNAs into training, validation and test sets to evaluate. Following gRNAde, we identify the structural clusters belonging to the RNAs which mainly includes riboswitches, aptamers, and ribozymes, and add all the RNAs in these clusters to the test set. The remaining clusters are randomly added to the training and validation splits.

For the complex dataset, we follow RNAflow and filter PDBBind to complexes where at least one protein $C_\alpha$ atom and RNA $C'4$ atom were within 7 Å, a threshold that has been used to perform alanine scans for protein-RNA interaction site analysis. For complexes containing many protein-RNA interaction sites, we use the interaction with least distance between the protein $C_\alpha$ atom and RNA $C'4$ atom. We follow RNAflow and filter to RNA chains of length [6,96], and protein chains are contiguously cropped to length 50. This results in 2k unique RNA sequences for which a total of 6233 structures are available. Further dataset statistics are available in Figure 3, and while there are many examples of short RNAs, the model is also exposed to longer RNA examples.

### F.2  SETTING DETAILS

For metrics, self-consistency RMSD (scRMSD) measures consistency between generated sequence-structure pairs by comparing the co-designed structure with the structure predicted from the generated sequence alone by the pretrained RF2NA model. TM-score evaluates global structural similarity, being less sensitive to local variations. LDDT calculates local structure confidence. For self-consistency Sequence recovery rate (scSeqRec), given co-designed 3D structure backbone, the pretrained RF2NA model recovers the sequence that folds into it and we measure the percentage of correctly recovered nucleotides in a co-designed sample sequence. For diversity metrics, structural diversity is quantified using pairwise RMSD distributions and qTM clustering with configurable

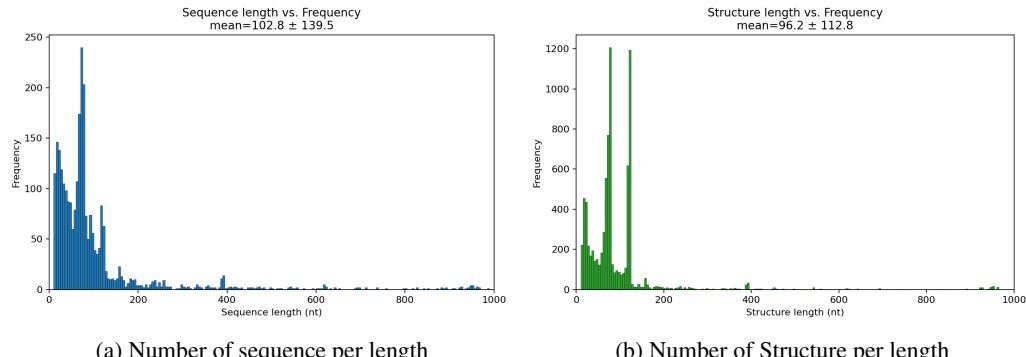

(a) Number of sequence per length      (b) Number of Structure per length

Figure 2: RNASolo data statistics. We plot histograms to visualize the diversity of RNAs available in terms of sequence length and number of structures per sequence.

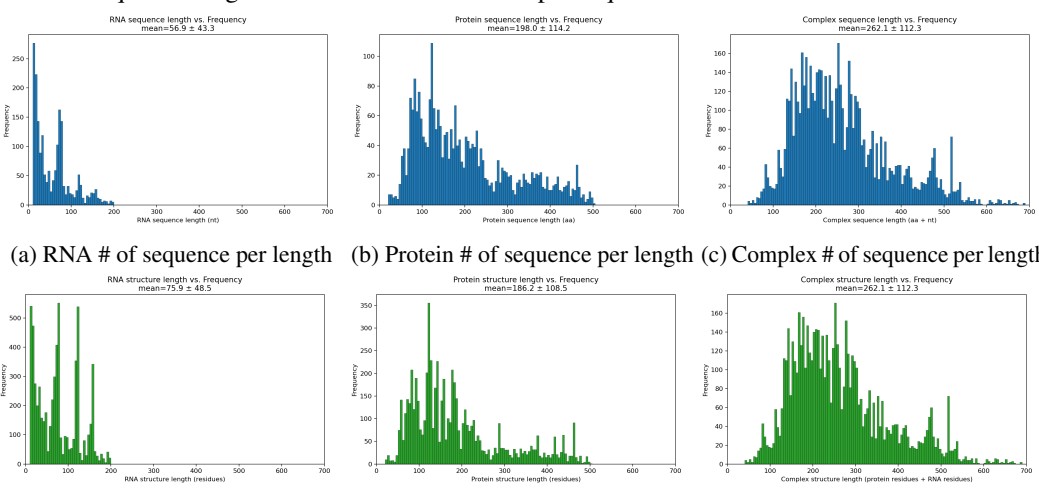

(a) RNA # of sequence per length   (b) Protein # of sequence per length   (c) Complex # of sequence per length

(d) RNA # of structure per length   (e) Protein # of structure per length   (f) Complex # of structure per length

Figure 3: RNA-protein complex data statistics. We plot histograms to visualize the diversity of RNAs, proteins, and complexes available in terms of sequence length and number of structures per sequence.

thresholds (TM=0.45). In addition, we report in the Appendix sequence diversity measured through unique sequence counts and secondary structure diversity analyzed through unique folding patterns. Regarding complex evaluation for conditional generation, ground truth RMSD (GT-RMSD) measures consistency between generated structure and condition by comparing the co-designed structure with the ground truth structure. Ground truth sequence recovery rate (GT-SeqRec) measures the percentage of correctly recovered nucleotides in the generated sequence comparing with the ground truth sequence. We also report in the Appendix binding confidence metrics iPTM and pTM scores from Chai-1 structure prediction. iPTM (interface predicted TM-score) measures the accuracy of predicted protein-RNA interface, while pTM (predicted TM-score) measures overall structural confidence.

## F.3 IMPLEMENTATION, SOFTWARE, AND HARDWARE

We initialize our model from pre-trained RoseTTAFold2NA weights and employ diffusion learning for joint sequence-structure generation.

Multiple sequence alignments (MSAs) are created for all protein and RNA sequences in the training and validation set. For MSA we make use of Protein databases UniRef30, BFD, and RNA databases Rfam18, RNAcentral17, and NT. Protein MSAs were generated in the same way as RoseTTAFold, using hhblits at successive E-value cutoffs, stopping when the MSA contains more than 10,000 unique sequences with >50% coverage. RNA MSAs are generated using a pareddown version of rMSA that removes secondary structure predictions. Sequences were searched using blastn30 over

three databases (RNAcentral17, rfam18 and nt) to first identify hits, then using nhmmer31 to rerank hits. We again use successive E-value cutoffs, stopping when the MSA contains more than 10,000 unique sequences with >50% coverage. MSA processing takes approximately 2 weeks on 320 CPUs for the two dataset.

Model configurations follow RF2NA, with 32 main blocks, 4 extra blocks and 4 reference block, MSA dimension 256, hidden dimension 32, pair dimension 128, 8 MSA heads, and 4 pair heads. For the diffusion process, we use 50 diffusion timesteps with a cosine noise schedule for continuous diffusion and linear schedule for discrete diffusion. Models are implemented in PyTorch, trained with Adam optimizer, cosine annealing schedules, and gradient accumulation for effective batch sizes. We employ mixed precision training for efficiency and implement early stopping based on validation scRMSD (or GT-RMSD for conditional task). For hyperparameters, the batch size is selected from {4, 8, 16, 32}, the learning rate from {5e-5, 1e-5, 1e-6}, the minimum learning rate from {1e-5, 1e-6, 1e-7}, and the maximum number of iterations from {15k, 20k, 30k}. $\lambda_{\text{seq}}\lambda_{\text{str}}$, $\lambda_{\text{rmsd}}$ are selected from {0.5, 1.0, 2.0, 3.0, 5.0, 6.0} and $\lambda_{\text{geom}}, \lambda_{\text{lj}}$ are selected from {0.03, 0.05, 0.1, 0.5, 1.0}. For computational efficiency, we limit training sequences to maximum length $L_{\text{max}} = 256$ nucleotides for RNA-only systems and RNA-protein complexes. Shorter sequences are padded for batch processing, while longer sequences serve as data augmentation, where a consecutive portion of $L_{\text{max}} = 256$ nucleotides within the sequence is randomly selected for the current batch. At inference we use 50 diffusion steps by default, with temperature scaling and partial-SVDD modes for refinement. For SVDD, we use $M = 10$ for practical efficiency, and set the temperature parameter $\alpha = 0.1$, since we primarily aim at reward refinement.

All models are trained on 4-8 NVIDIA A100 or A6000 GPUs. The model converges after approximately 10k-15k gradient steps, requiring 2.25 seconds per iteration of training on our hardware configuration. With our validation and early stopping, this results in 10 hours of training.

### F.4 BASELINE DETAILS

We compare with existing RNA co-design methods MMDiff and RiboGen. For MMDiff, we follow and adapt its official code implementation under our settings and datasets. For RiboGen, due to lack of code, implementation detail or numerical results, we perform visual comparison in Appendix G.2. A random generation baseline is also implemented by random model initialization. For protein-conditioned design, we compare to existing conditional co-design method RNAFlow (Nori & Jin, 2024). We run RNAFlow-Base, which is the only implementation presented in the official code. We follow RNAFlow's dataset splits, and use its published checkpoint and code implementation to produce the results. Note that we calculate and report all-atom RMSD instead of backbone-only RMSD in RNAFlow's original implementation. Following prior work, we sample 10 RNA designs per complex in the test set, conditioning on the protein backbone and sequence. Note that our method does not post-hoc "sequence" a generated backbone; instead, it co-designs sequence and structure within one diffusion trajectory. For the baseline that first designs backbones with RNA-FrameFlow and then applies gRNAde to generate sequences for those backbones, to align with our evaluation pipeline for fair comparison, we use RF2NA for forward-folding instead of RhoFold to calculate the reference structure. For RNA-FrameFlow model, we follow their paper and use 6 IPA blocks with 3 torsion predictor layers, and $N_T = 50$. We adapt the architecture and implementation from their official code, and run training till convergence. Note that the scRMSD/scTM etc. is calculated only on all backbone atoms given the setting difference of RNA-FrameFlow. Also, scSeqRec under this setting is essentially an evaluation of gRNAde aligned to RF2NA in inverse folding.

Regarding sequence generator + folding tool baselines, we believe it is a less informative and somewhat misleading baseline for our evaluation protocol. Current evaluation in this field already uses external structure predictors as a common oracle. All recent joint-generation and inverse-folding works (including gRNAde, MMDiff, RNAFlow, and related protein-design methods) evaluate generated molecules by folding them with a strong external predictor and then computing metrics on the folded structures. In other words, the structure predictor is part of the evaluation pipeline, not the generative model itself. Our work follows this established protocol. Thus a sequence generator + folding pipeline conflates the generation quality with oracle power. If we treat it as a competing method, we are no longer comparing generative models under the same evaluation oracle, instead we are comparing composite systems where RF2NA now acts as a powerful optimizer inside the baseline. This makes it difficult to attribute performance differences to the generative model versus

Table 5: Results of more self-consistency and diversity metrics on single RNA, RNA-protein complex, and conditional RNA design (RF2NA pre-training split) tasks. Average value and standard deviation are reported for all metrics.

| Method | scRMSD (Å ↓) | scTM-score (↑) | LDDT (↑) | scSeqRec (% ↑) | qTMclust Diversity (↑) | Sequence Diversity (↑) | Secondary Struc Diversity (↑) |
|---|---|---|---|---|---|---|---|
| Single RNA | **3.43**±0.51 | **0.71**±0.04 | **0.74**±0.06 | **48.57**±4.20 | **1.00**±0.00 | **1.00**±0.00 | **1.00**±0.00 |
| Complex | **7.43**±0.88 | **0.42**±0.07 | **0.71**±0.05 | **52.91**±3.90 | **1.00**±0.00 | **1.00**±0.00 | **0.94**±0.07 |
| Conditional | **10.06**±1.57 | **0.29**±0.13 | **0.70**±0.05 | **62.23**±4.68 | **1.00**±0.00 | **1.00**±0.00 | **0.96**±0.05 |

Table 6: Chai-1 confidence metrics on protein-conditioned RNA design. "GT data" are max-length-truncated complexes drawn from the training distribution and used as a reference; RiboDiff results are our conditional generations, and SVDD results applies our inference-time resampling. Mean ± standard deviation across targets.

| Metric | GT data | RiboDiff | RiboDiff + SVDD (on iPTM) |
|---|---|---|---|
| iPTM | $0.244 \pm 0.113$ | $0.166 \pm 0.060$ | $0.187 \pm 0.084$ |
| pTM | $0.552 \pm 0.090$ | $0.520 \pm 0.095$ | $0.509 \pm 0.076$ |

the folding oracle. For these reasons, we view it as a different problem formulation, and not directly comparable to our joint sequence–structure models under the standard evaluation protocol. To keep the comparison focused and interpretable, we therefore prioritize joint/inverse-folding baselines such as MMDiff and RNAFlow/gRNAde-style pipelines.

# G  FURTHER EXPERIMENTAL RESULTS

## G.1  RESULTS ON FURTHER METRICS

Table 5 reports results of more self-consistency and diversity metrics on single RNA, RNA-protein complex, and conditional RNA design. For Single-RNA co-design, all three diversity measures are 1.00, indicating no mode collapse and that generated sequences, global 3D folds (qTMclust at TMcutoff 0.45), and secondary structures span the design space. For RNA-protein complexes, diversity stays maximal overall, with a small dip in secondary-structure diversity, consistent with reuse of interface-friendly motifs (stems capped by short loops/bulges). For protein-conditioned RNA design, RNA must both fold and complement a fixed protein, and global scRMSD stands larger, while local geometry is still preserved. The performance in scSeqRec suggests that under strong conditioning, the joint model produces backbones whose compatible sequences are highly constrained and thus easier to recover, reflecting tight sequence-structure-interface coupling. Diversity remains high, and secondary-structure diversity suggests modest convergence toward a subset of secondary topologies that pack well against proteins.

Table 6 reports Chai-1 confidence metrics on protein-conditioned RNA design, which are more explicit metrics regarding binding affinity compared to GT-RMSD. The pTM scores close to the ground-truth indicate that our designed RNAs form internally coherent folds when conditioned on the protein, suggesting the model preserves the RNA's global topology reasonably well. The iPTM scores, which emphasize protein–RNA interface quality, are lower than the truncated ground-truth, as expected for de novo designs. Importantly, value-based importance sampling improves iPTM by 12.7%, partially closing the gap to ground-truth and indicating that the resampling step successfully biases generations toward better binding complementarity. To push interface quality further without sacrificing fold stability, training-time objectives could explicitly weight interface contacts (e.g., clash penalties + contact rewards, interface-conditioned losses), or we could combine resampling with short interface-focused refinement before final scoring.

A risk in the unconditional setting is memorization, that the model might simply reproduce training structures instead of learning a broad prior. To address this, we add experiments on another metric, novelty, where we compute compute the TM-score between all pairs of generated samples and structures in our training set, and if the highest align for a generated sample is <0.45, it is considered novel. As shown in Table 7, we achieve reasonably high novelty, suggesting that the model is not simply memorizing the training structures.

Table 7: Novelty comparison across methods.

| Methods | Random generation | MMDIFF | RNA-FrameFlow + gRNAde | RiboDiff |
|---------|-------------------|--------|------------------------|----------|
| Novelty | 100.0% | 82.6% | 55.8% | 76.0% |

Table 8: Stereochemical validity metrics for generated single-RNA structures.

| Method | Clashscore ($\downarrow$) | Bond/angle outliers (% $\downarrow$) | Backbone conformer outliers (% $\downarrow$) |
|--------|----------|----------------------|------------------------------|
| Random generation | 45.2 | 6.8 | 18.5 |
| MMDIFF | 19.7 | 4.9 | 14.2 |
| RiboDiff w/o $L_{geom}$ | 15.6 | 2.4 | 7.9 |
| RiboDiff w/o $L_{lj}$ | 13.8 | 2.2 | 7.1 |
| RiboDiff | **9.4** | **1.6** | **5.3** |

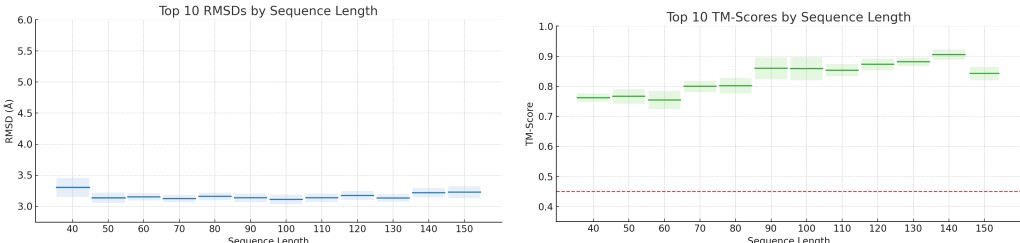

(a) Top-10 scRMSD per length (mean $\pm$ std band)  (b) Top-10 scTM-score per length (mean $\pm$ std band)

Figure 4: scRMSD and scTM-score across different sequence lengths (40-150 nucleotides), showing the top 10 generated structures for each length, in consistent with RiboGen (Rubin et al., 2025).

To strengthen external validation, for unconditional RNA design task, we add physical plausibility metrics. We explicitly evaluate whether the generated RNA structures are physically plausible as standalone 3D models, which is exactly what our geometric and Lennard-Jones losses are designed to enforce. Following standard practice in RNA-Puzzles and CASP RNA assessments, we compute MolProbity-style clashscore and the fraction of stereochemical outliers (bond/angle outliers and backbone conformer outliers) on generated single-RNA structures. As shown in Table 8, RiboDiff achieves the lowest clashscore and the fewest stereochemical outliers. These results support that RiboDiff learns a physically valid prior over RNA structures precisely through the designed terms.

## G.2 COMPARISON WITH RIBOGEN

For RiboGen, due to lack of code, implementation detail or numerical results, we perform visual comparison here. As shown in Figure 4, we report results on 10 best-performing samples per length. Our results demonstrate significant improvements over RiboGen (Rubin et al., 2025), suggesting stable self consistency and generalization across different lengths. The top-10 scRMSD by length indicates the model can reliably produce very accurate self-consistent designs across lengths, with only modest degradation as structures grow. The top-10 scTM-score suggests the best samples achieve increasingly correct global topology as helices lengthen (TM is less penalized by local loop deviations), while the slight drop at the extreme end likely reflects edge effects and greater conformational heterogeneity.

## H ABLATION STUDIES

As ablation studies of our joint diffusion, we compare the performances under several adjusted settings. First, we study the case where pre-trained priors are not used. Next, we adjust the training loop to allow only diffuse sequence (keeping structure clean) or only diffuse structure (keeping sequence clean) and use flexible loss function that can handle sequence-only, structure-only, or joint diffusion modes. We study the cases of "alternating diffusion" between sequence- and structure-only.

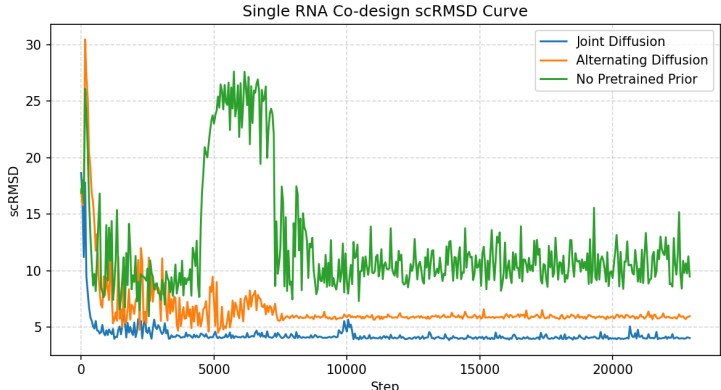

Figure 5: scRMSD curve by learning steps regarding our joint diffusion (with pre-trained priors), alternating sequence- and structure-only diffusion, and joint diffusion without pre-trained priors.

Table 9: Ablation of loss terms on single RNA design.

| Method | Success rate (% ↑) | scRMSD (Å ↓) | scTM-score (↑) | LDDT (↑) | qTMclust Diversity (↑) |
|---|---|---|---|---|---|
| w/o $L_{\text{geom}}$ | 92.77 | 4.94 | 0.69 | 0.71 | 1.00 |
| w/o $L_{\text{lj}}$ | 95.30 | 4.67 | 0.69 | 0.72 | 0.94 |
| w/o $L_{\text{str}}$ | 58.64 | 5.85 | 0.67 | 0.68 | 0.83 |
| w/o $L_{\text{rmsd}}$ | 30.08 | 16.72 | 0.39 | 0.48 | 0.68 |
| Full loss | **97.38** | **3.43** | **0.71** | **0.74** | **1.00** |

As shown in Figure 5, joint diffusion learns scRMSD to 4 Å within the first few hundred steps, while alternating diffusion improves but stalls higher, never matching the performance. Training without pre-trained priors is markedly unstable: after an initial drop it exhibits a long plateau with high variance before partially recovering. The gap quantifies the value of co-denoising sequence and structure together that when both channels evolve jointly, cross-modal constraints are enforced at every step, yielding tighter sequence–structure compatibility (lower scRMSD). Alternating sequence-only and structure-only phases weakens this coupling. Each phase can undo progress made by the other, so the model converges to a looser equilibrium. The no-pretrain trace highlights the role of pretrained structural priors (RF2NA) as a strong inductive bias, without which optimization is noise-sensitive, prone to bad minima, and slow to stabilize, especially in the data-scarce RNA regime.

Our current loss combines sequence cross-entropy, FAPE, RMSD, geometric penalties, and a Lennard-Jones clash term. To better show the effect of each component, we add an ablation table on single RNA co-design task comparing settings without certain components, and report performances metrics. As shown in Table 9, removing the global RMSD term has the most dramatic effect, confirming that $l_rmsd$ is critical for enforcing globally correct RNA folds rather than merely local improvements. Removing the structure-alignment loss also significantly hurts performance, indicating that the RF2NA-guided structural prior is essential for robust sequence-structure co-consistency and maintaining diverse yet correct conformations. In contrast, dropping the auxiliary geometric and Lennard-Jones terms yields more moderate degradation. These results support our design, with the rmsd and str terms providing the bone of global correctness, and the geometric and steric losses further sharpening physical plausibility and stability.

We also implement a "RF2NA-backbone + gRNAde" study, where we adjust our diffusion design to train only a backbone structure diffusion and then apply gRNAde for inverse folding. We evaluate this pipeline on our single RNA co-design task, under the same train/test splits and metrics. Note that the scRMSD/scTM etc. is calculated only on all backbone atoms given the setting difference. As shown in Table 10, the results remain far behind RiboDiff's performances. These results support our claim that joint sequence-structure co-diffusion provides stronger sequence–structure self-consistency.

To validate that the complex conditioning is warranted, we add the following ablations in the protein-conditioned design experiments. We compare the current conditioning with sequence-only

Table 10: Backbone-only RF2NA + gRNAde pipeline versus RiboDiff on single RNA design.

| Method | Success rate (% ↑) | scRMSD (Å ↓) | scTM-score (↑) | LDDT (↑) | scSeqRec (% ↑) | qTMclust Diversity (↑) |
|---|---|---|---|---|---|---|
| RF2NA-backbone + gRNAde (Backbone) | 19.34 | 17.59 | 0.34 | 0.42 | 45.50 | 0.80 |
| RiboDiff | **97.38** | **3.43** | **0.71** | **0.74** | **48.57** | **1.00** |

Table 11: Conditioning ablations on the RF2NA Pre-Training Split for protein-conditioned RNA design.

| Method (RF2NA Pre-Training Split) | GT-RMSD (Å ↓) | GT-SeqRec (% ↑) |
|---|---|---|
| Sequence-only conditioning | 16.2 | 33.5 |
| Structure-only conditioning | 15.1 | 45.9 |
| Modular conditioning baseline | 15.5 | 38.2 |
| Full RF2NA conditioning (ours) | **13.2** | **56.3** |

conditioning (supply only the protein primary sequence, with backbone coordinates replaced by a generic template) and structure-only conditioning (supply only the protein structure with masked protein sequence tokens in the tracks) and evaluate the trained models on GT-RMSD and GT-SeqRec metrics. Also, to mimic a simpler approach, we implement a baseline where the protein is encoded by a ESM-IF model into residue embeddings, and these embeddings are then injected only as additional "conditioning tokens" into the RNA sequence/structure tracks, without fully coupling via RF2NA's pair track. This variant is closer in spirit to typical conditional diffusion setups used in images and text. On the RF2NA Pre-Training Split for protein-conditioned RNA design, as shown in Table 11, using only protein sequence or only protein structure as conditioning degrades performance relative to full conditioning. This indicates that sequence or structure alone is not sufficient to capture the detailed protein-RNA interface. The modular conditioning baseline, also lags behind fully coupled RF2NA conditioning. Although various modular conditioning strategies are to be tested, these results can directly support our design choice that complex coupling provides measurable benefits.

# I VISUALIZATION

Figure 6 show visualization of generated RNA structures using RiboDiff on single RNA and protein-conditioned RNA co-design tasks. for single-RNA co-design, the orange (designed) traces follow the green self-folded references closely along the helical stems. Base-paired segments superpose with small axial offsets and only mild twist differences. Deviations are concentrated at loop apices and helix–helix junctions, where the orange backbone is slightly inflated relative to the reference path. This spatial is exactly what we expect from a model that captures the dominant A-form geometry and stacking while leaving flexible tertiary details underconstrained. For protein-conditioned design, the designed RNA docks into protein groove with the stem axis and sugar–phosphate backbone. The overlap of orange and green within the contact region suggests that key interface geometry is recovered, with correct placement of the principal helical elements and a largely correct binding pose. Physically plausible RNA geometry can be observed, although the tertiary placement and long-range contact network are not fully captured. This is typical for the setting when the interface is shallow or ambiguous, multiple binding poses exist, or long loops mediate docking, pointing to the need for stronger interface-aware guidance.

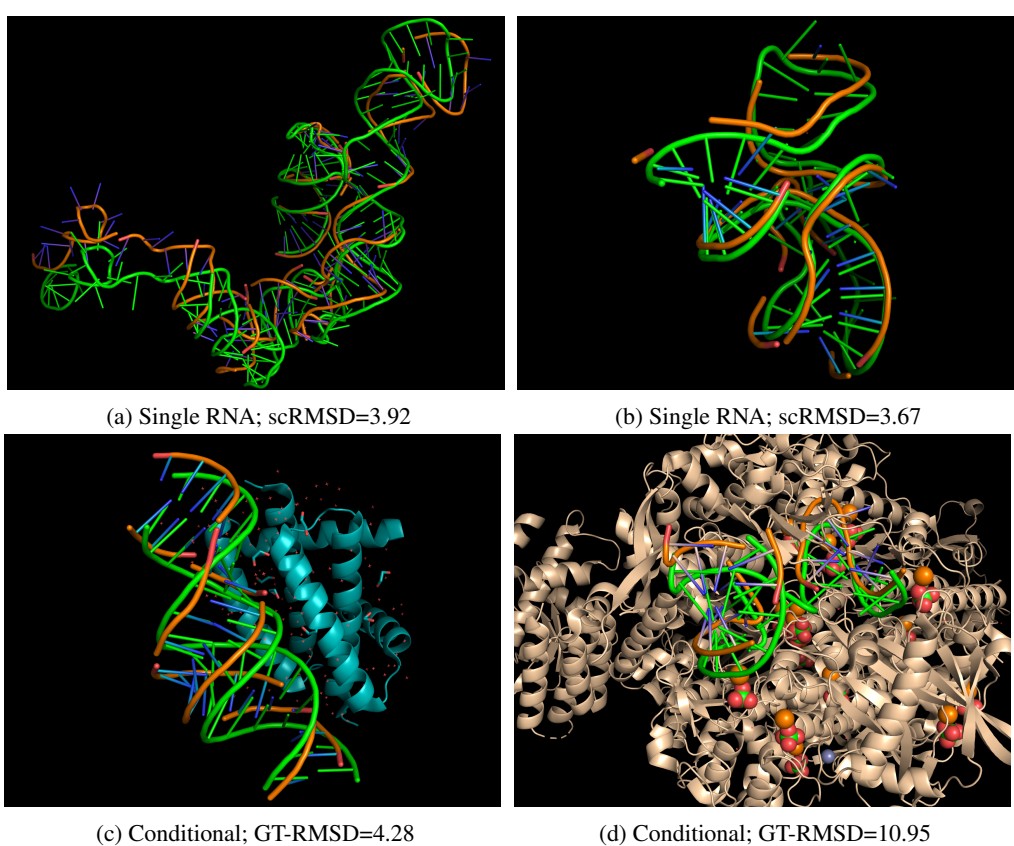

(a) Single RNA; scRMSD=3.92        (b) Single RNA; scRMSD=3.67

(c) Conditional; GT-RMSD=4.28        (d) Conditional; GT-RMSD=10.95

Figure 6: Visualization of generated RNA structures using RiboDiff on single RNA and protein-conditioned RNA co-design tasks. For single RNA, (orange) colored RNA are co-designed structures while green RNA are reference structures generated by pretrained model. For protein-conditioned RNA, ribbon structures are protein conditions, (orange) colored RNA are co-designed structures while green RNA are ground truth reference structures in the complex.

