# OpenReview forum: "A Joint Diffusion Model with Pre-Trained Priors for RNA Sequence-Structure Co-Design"
_ICLR.cc/2026/Conference — ICLR 2026 Poster_

### Official Review · Reviewer_Anrp · 2025-10-31

**Soundness:** 2
**Presentation:** 2
**Contribution:** 2
**Rating:** 4
**Confidence:** 3

**Summary:**

The paper introduces a novel joint RNA sequence and structure denoising model that integrates the RoseTTAFold2NA structural prediction network as the denoiser within a generative diffusion framework. This approach is designed to predict RNA sequences under various conditioning settings, including protein-guided design. The central motivation is to leverage the strong inductive bias of a pretrained structural model (RoseTTAFold2NA) to overcome the significant data scarcity, particularly for RNA sequences with paired structural annotations.

**Strengths:**

1. The innovative use of the pretrained RoseTTAFold2NA model as the core denoiser provides a powerful structural inductive bias. This design choice is highly effective in mitigating the challenges posed by the limited availability of high-quality, structure-annotated RNA data, thereby enhancing the model's generative capacity.

2. The method extends beyond existing structural diffusion models (like RFdiffusion) by integrating modern conditional guidance to explicitly incorporate protein sequence and/or structure information.

**Weaknesses:**

1. The primary evaluation relies heavily on self-consistency metrics (scTM, scRMSD, sc-lDDT), which measure the agreement between the generated structure and the structure predicted from the generated sequence. While valuable for checking folding consistency, these internal metrics do not constitute external validation against true ground-truth structures or de novo design goals.

2. The work employs highly complex, high-dimensional inputs (full protein sequence and backbone coordinates) as conditioning signals injected directly into the main backbone. Unlike simpler conditioning approaches, the efficacy of using such complex conditioning without dedicated analysis is questionable. A deeper justification or ablation is needed to validate that this complex, integrated injection mechanism is superior to simpler or more modular conditioning strategies.

**Questions:**

The content in this section is identical to the points raised in Weaknesses. I would be very willing to increase my score if the authors successfully address these concerns.

---

> ### Author Response · Authors · 2025-11-25
> **Response to Reviewer Anrp**
>
> Dear Reviewer Anrp,
>
> Thank you for your constructive suggestions and insightful comments! We have made efforts to thoroughly improve our work accordingly and provide responses for each concern here. Please also check the revised paper and Appendices for additional experiments and clarifications.
>
> > The primary evaluation relies heavily on self-consistency metrics (scTM, scRMSD, sc-lDDT), which measure the agreement between the generated structure and the structure predicted from the generated sequence. While valuable for checking folding consistency, these internal metrics do not constitute external validation against true ground-truth structures or de novo design goals.
>
>
> We appreciate the reviewer raising this point. We agree that self-consistency metrics alone primarily check whether the sequence and structure we co-generate are validly coherent. We would like to clarify what is already in the paper and then describe additional evaluations we add.
> - For conditional design tasks, we **already include metrics like ground-truth RMSD (GT-RMSD) and ground-truth sequence recovery (GT-SeqRec)** where the generated RNA is compared directly to the experimental RNA structure in the complex. Since references exist for conditional generation, these GT metrics are reported alongside self-consistency metrics and show that our model improves not only internal consistency but also actual deviation from experimentally determined structures in the conditional setting.
> - In the de novo RNA/complex design setting, the tasks are unconditional generation and by construction there is **no ground-truth structure** for a designed RNA. The goal is to produce sequences that fold into stable, well-defined shapes.
> - Self-consistency metrics (scTM, scRMSD, sc-lDDT, scSeqRec) are thus standard in RNA/protein generative works as proxy objectives. These metrics are particularly useful here because with a **strong multi-context structure predictor as the evaluator, the reference structure from the evaluator is considered valid**. If a sequence re-folds into a reference structure close to the one we generated, it is strong evidence that the generated structure is valid in all-atom structure, as well as encoding foldable sequence–structure relationships.
>
> |Method|Clashscore (↓)|Bond/angle outliers (%↓)|Backbone conformer outliers (%↓)|
> |-|-|-|-|
> |Random generation|45.2|6.8|18.5|
> |MMDIFF|19.7|4.9|14.2|
> |RiboDiff w/o L_geom|15.6|2.4|7.9|
> |RiboDiff w/o L_lj|13.8|2.2|7.1|
> |RiboDiff|**9.4**|**1.6**|**5.3**|
>
> - To address the reviewer’s concern and strengthen external validation, for unconditional RNA design task, **we add physical plausibility metrics**. We explicitly evaluate whether the generated RNA structures are physically plausible as standalone 3D models, which is **exactly what our geometric and Lennard–Jones losses are designed to enforce**. Following standard practice in RNA-Puzzles and CASP RNA assessments, we compute MolProbity-style clashscore and the fraction of stereochemical outliers (bond/angle outliers and backbone conformer outliers) on generated single-RNA structures. As shown in the above Table, RiboDiff achieves the lowest clashscore and the fewest stereochemical outliers. These results support that RiboDiff learns a physically valid prior over RNA structures precisely through the designed terms.
>
> > The work employs highly complex, high-dimensional inputs (full protein sequence and backbone coordinates) as conditioning signals injected directly into the main backbone. Unlike simpler conditioning approaches, the efficacy of using such complex conditioning without dedicated analysis is questionable. A deeper justification or ablation is needed to validate that this complex, integrated injection mechanism is superior to simpler or more modular conditioning strategies.
>
>  We appreciate this point and agree that conditioning mechanisms can be justified and ablated.

---

> > ### Author Response · Authors · 2025-11-25
> > **Response to Reviewer Anrp (II)**
> >
> > - Our design choices are driven by RF2NA’s architecture and by the nature of RNA–protein interfaces.
> >   - RF2NA is a multi-context structure predictor that is designed to take as input full protein and nucleic acid simultaneously. Its three-track architecture is built to model cross-chain, residue-level interactions. By feeding the entire protein chain into RF2NA and diffusing only the RNA chain, we allow the model to exploit residue-level geometric constraints at the interface, long-range protein context that influences where and how RNA binds and learned priors on protein–RNA contacts encoded in the pair track.
> >   - Simpler conditioning (e.g., a global protein embedding or interface-only patch) risks discarding information that RF2NA already knows how to use for precise interface geometry.
> >   - Empirically, in the protein-conditioned binder design task, our method outperforms RNAFlow, which uses a **modular setup**, with a GNN for sequence generation conditioned on protein structure plus RF2NA only as an external structure predictor. Our gains in GT-RMSD, ipTM-like interface scores, and self-consistency indicate that the tightly integrated conditioning through RF2NA’s backbone is working as intended.
> >
> > |Method (RF2NA Pre-Training Split)|GT-RMSD (Å↓)|GT-SeqRec (%↑)|
> > |-|-|-|
> > |Sequence-only conditioning|16.2|33.5|
> > |Structure-only conditioning|15.1|45.9|
> > |Modular conditioning baseline|15.5|38.2|
> > |Full RF2NA conditioning (ours)|**13.2**|**56.3**|
> >
> > - To directly address the reviewer’s concern and validate that the complex conditioning is warranted, we **add the following ablations** in the protein-conditioned design experiments.
> >   - We compare the current conditioning with sequence-only conditioning (supply only the protein primary sequence, with backbone coordinates replaced by a generic template) and structure-only conditioning (supply only the protein structure with masked protein sequence tokens in the tracks) and evaluate the trained models on GT-RMSD and GT-SeqRec metrics.
> >   - Also, to mimic a **simpler approach**, we implement a baseline where the protein is encoded by a ESM-IF model into residue embeddings, and these embeddings are then injected only as additional “conditioning” into the RNA sequence/structure tracks, without fully coupling via RF2NA’s pair track. This variant is closer in spirit to typical conditional diffusion setups used in images and text.
> >   - On the RF2NA Pre-Training Split for protein-conditioned RNA design, as shown in the Table above, using only protein sequence or only protein structure as conditioning degrades performance relative to full conditioning. This indicates that sequence or structure alone is not sufficient to capture the detailed protein–RNA interface. The modular conditioning baseline, also lags behind fully coupled RF2NA conditioning. Although various modular conditioning strategies are to be tested, these results can directly support our design choice that complex coupling provides measurable benefits. We also include these ablations in the paper revision.
> >
> >
> > We sincerely thank you for your time! Hope we have addressed your concerns through practical efforts and shown the contributions and significance of our work. We look forward to your reply and further discussions, thanks!
> >
> > Sincerely,
> >
> > Authors

---

### Official Review · Reviewer_vJj9 · 2025-10-31

**Soundness:** 3
**Presentation:** 3
**Contribution:** 2
**Rating:** 6
**Confidence:** 5

**Summary:**

This paper proposes a joint diffusion model for RNA sequence–structure co-design, embedding the pre-trained RoseTTAFold2NA as a denoiser within dual diffusion processes: a discrete diffusion for RNA sequences and an SE(3)-equivariant diffusion for 3D structures. The method enables simultaneous generation of sequences and structures and supports conditional tasks such as RNA–protein complex or protein-conditioned RNA design. Experiments show solid improvements over previous RNA-specific diffusion or flow models trained from scratch, demonstrating that leveraging pre-trained structural priors can improve sample efficiency and accuracy in data-scarce RNA settings.

**Strengths:**

The work is technically sound and well-executed, with clear formulation and comprehensive evaluation. Integrating pre-trained structural knowledge from RoseTTAFold2NA into a joint diffusion framework is a reasonable and effective strategy for improving RNA generative design. The paper is well-written, the methodology is consistent with recent trends in biomolecular generation, and the results are convincing and reproducible. Overall, the paper completes a meaningful and well-defined task without major flaws.

**Weaknesses:**

The novelty of the approach is limited. The overall framework closely parallels prior work in protein design, particularly “Generative Flows on Discrete State-Spaces: Enabling Multimodal Flows with Applications to Protein Co-Design”, which introduced a similar idea of coupling discrete and continuous generative processes for sequence–structure modeling. The current paper largely adapts this concept to the RNA setting, without introducing substantial methodological innovation or theoretical insight beyond that extension. While the adaptation is meaningful, the contribution is incremental rather than conceptually new.

**Questions:**

None

---

> ### Author Response · Authors · 2025-11-25
> **Response to Reviewer vJj9**
>
> Dear Reviewer vJj9,
>
> Thank you for your acknowledgment of our work and insightful comments! We have made efforts to thoroughly improve our work accordingly and provide responses for each concern here. Please also check the revised paper and Appendices for additional experiments and clarifications.
>
> > The novelty of the approach is limited. The overall framework closely parallels prior work in protein design, particularly “Generative Flows on Discrete State-Spaces: Enabling Multimodal Flows with Applications to Protein Co-Design”, which introduced a similar idea of coupling discrete and continuous generative processes for sequence–structure modeling. The current paper largely adapts this concept to the RNA setting, without introducing substantial methodological innovation or theoretical insight beyond that extension. While the adaptation is meaningful, the contribution is incremental rather than conceptually new.
>
>
> We appreciate the reviewer drawing the connection to DFM-Multiflow. Our work is indeed inspired by the general idea of joint sequence–structure generative modeling, since co-design is an important task drawing increasing attention in many domains. However, our framework differs from the DFM work in several important ways, both methodologically and in scope, beyond a straightforward adaptation to RNA.
> - **Diffusion with RF2NA priors vs. flow from-scratch**
>   - The DFM-Multiflow work learns a DFM for the sequence and a continuous-time flow model, FrameFlow, for protein structures, **trained from scratch**.
>   - Our work uses a dual diffusion process (discrete diffusion on nucleotides + SE(3)-equivariant diffusion on rigid frames) with RF2NA embedded as the pretrained denoiser prior. This yields a pre-trained multi-context trunk capable of handling proteins, RNAs, DNA, and complexes, and a diffusion parameterization that explicitly matches RF2NA’s rigid-frame representation and internal three-track structure. This **“pretrained predictor as denoiser” formulation** and the way we adapt RF2NA’s architecture for generative RNA (and complex/conditional) co-design is substantially different from DFM-Multiflow’s from-scratch protein flow model.
>   - The DFM-Multiflow work mainly positions with the insight that discrete flow-based model can be realized using continuous time markov chains, and use one DFM and one FrameFlow model to form Multiflow, which is a different focus from our paper. We embeds RF2NA as the denoiser into a dual diffusion model with the insight that **injecting rich cross-molecular priors while enabling sample-efficient learning from limited RNA data**.
>
> - DFM-Multiflow focuses on **single protein co-design**. In contrast, our framework exploits RF2NA’s multi-context backbone to support three tasks in the framework, single RNA design, **RNA–protein complex design, and Protein-conditioned RNA binder design**. This cross-molecular generative scope is, not covered by DFM-Multiflow.
> - We incorporate **RL-style guidance** (value-based decoding) into the diffusion sampling process, using task-aligned non-differentiable rewards such as interface quality, and ipTM-like scores. DFM-Multiflow does not perform reward-guided inference on design objectives.
> - RNA generative modeling presents challenges not handled in DFM-Multiflow. Our model jointly diffuses all-atom coordinates and discrete nucleotides, together with geometric and steric losses (e.g., Lennard–Jones) tailored to RNA. On the empirical side, we show that integrating RF2NA into a dual diffusion framework yields significant gains over joint RNA diffusion models trained from scratch and two-stage pipelines. This systematically demonstrates, for the first time, that pretraining-guided joint diffusion over RF2NA is a powerful and practical recipe for RNA design under data scarcity.
>
> We agree that the high-level conceptual template of joint discrete + continuous generative modeling is related to DFM-Multiflow. We make this relationship more explicit in Related Work of the paper revision, while also clarifying the above concrete differences in architecture, training regime, task scope, and RL-guided inference.
>
> We sincerely thank you for your time! Hope we have addressed your concerns through practical efforts and shown the contributions and significance of our work. We look forward to your reply and further discussions, thanks!
>
> Sincerely,
>
> Authors

---

### Official Review · Reviewer_9N8E · 2025-10-31

**Soundness:** 2
**Presentation:** 3
**Contribution:** 2
**Rating:** 4
**Confidence:** 3

**Summary:**

The authors propose a new method for RNA structure generation that utilises RF2NA as a pretrained structure encoder and train a generative model on top of this, similar to how RFDiffusion was trained for proteins.

**Strengths:**

[S1]  The authors use a pretrained encoder for their diffusion process and demonstrate via ablations that this is a strong contributor to their performance.

[S2] The authors do not investigate only RNA generation in isolation, but also test RNA-Protein complex design, a more practically relevant task.

**Weaknesses:**

[W1] Missing baselines: the authors describe RNA-FrameFlow and gRNAde as a RNA backbone structure generation and RNA inverse folding model, but do not compare to it empirically. Given that in that paper the authors show that their method outperforms MMDiff (the main baseline in this paper), the authors should compare to it.

[W2] While the authors perform an ablation study with respect to their pre-trained prior, their framework contains a lot more components they claim are important, for example RL-enhanced diffusion inference, various auxiliary losses and their codiffusion objective. More systematic ablations here would strengthen the paper, for example 1) what happens without these auxiliary losses, or 2) what happens if one trains their model with just backbone design and then adds gRNAde on top instead of codiffusion?

**Questions:**

[Q1] Recent all atom structure prediction models unify proteins rna etc into a consistent framework instead of having specialised structure predictions methods like RF2NA. Do you think something similar is realistic/desirable in design?

---

> ### Author Response · Authors · 2025-11-25
> **Response to Reviewer 9N8E**
>
> Dear Reviewer 9N8E,
>
> Thank you for your constructive suggestions and insightful comments! We have made efforts to thoroughly improve our work accordingly and provide responses for each concern here. Please also check the revised paper and Appendices for additional experiments and clarifications.
>
> > [W1] Missing baselines: the authors describe RNA-FrameFlow and gRNAde as a RNA backbone structure generation and RNA inverse folding model, but do not compare to it empirically. Given that in that paper the authors show that their method outperforms MMDiff (the main baseline in this paper), the authors should compare to it.
>
> We thank the reviewer for emphasizing the importance of broader baselines.
> - Our current choice of baselines was guided by the goal of comparing between joint sequence–structure modeling. For single RNA and complex co-design, MMDiff is (to our knowledge) the only publicly available joint diffusion model for RNA and RNA–protein complexes, and is therefore the most direct baseline. We also compared with RiboGen in the Appendix. For protein-conditioned design, we compare against RNAFlow, which represents a joint pipeline that conditions on a protein.
> |Method|Success rate (%↑)|scRMSD (Å↓)|scTM-score (↑)|LDDT (↑)|scSeqRec (%↑)|qTMclust Diversity (↑)|
> |-|-|-|-|-|-|-|
> |Random generation|0.00±0.00|39.74±4.82|0.05±0.03|0.23±0.05|1.06±0.40|0.99±0.01|
> |MMDIFF|8.86±3.12|35.77±5.15|0.12±0.06|0.33±0.07|23.90±8.42|**1.00±0.00**|
> |RNA-FrameFlow + gRNAde (Backbone)|15.52±4.33|18.65±4.27|0.32±0.08|0.43±0.12|45.65±2.23|0.76±0.10|
> |RiboDiff|**97.38±4.86**|**3.43±0.51**|**0.71±0.04**|**0.74±0.06**|**48.57±4.20**|**1.00±0.00**|
>
> - That said, we agree that evaluating two-stage pipelines like structure generator + inverse folding will further clarify the value of joint co-design.
>   - We have **added a baseline that first designs backbones with RNA-FrameFlow and then applies gRNAde to generate sequences for those backbones**, closely mirroring the pipeline described in RNA-FrameFlow paper. To align with our evaluation pipeline for fair comparison, we use RF2NA for forward-folding instead of RhoFold to calculate the reference structure.
>   - For RNA-FrameFlow model, we follow their paper and use 6 IPA blocks with 3 torsion predictor layers, and $N_T = 50$. We adapt the architecture and implementation from their official code, and run training till convergence. We evaluate this pipeline on our single RNA co-design task, under the same train/test splits and metrics (scRMSD, scTM, lDDT, scSeqRec, qTMclust) as in Tables 1–2. Note that the scRMSD/scTM etc. is calculated only on all backbone atoms given the setting difference of RNA-FrameFlow. Also, scSeqRec under this setting is essentially an evaluation of gRNAde aligned to RF2NA in inverse folding.
>   - This directly address whether “backbone-then-sequence” pipelines can match the self-consistency and sequence-realizability of our joint co-design. As shown in the Table above, RiboDiff’s performances remain strong compared with this two-stage pipeline. These results support our claim that joint sequence–structure co-diffusion with a pretrained RF2NA prior provides stronger and more diverse sequence–structure self-consistency than a backbone-then-inverse-folding pipeline, particularly under structure metrics that involve refolding from sequence. We have added these baseline results in the revised paper and explicitly discuss their strengths and limitations.
>
> > [W2] While the authors perform an ablation study with respect to their pre-trained prior, their framework contains a lot more components they claim are important, for example RL-enhanced diffusion inference, various auxiliary losses and their codiffusion objective. More systematic ablations here would strengthen the paper, for example 1) what happens without these auxiliary losses, or 2) what happens if one trains their model with just backbone design and then adds gRNAde on top instead of codiffusion?
>
> We thank the reviewer for this point and agree that additional ablations will make the contributions clearer.
> - RL-enhanced inference
>   - Table 4 already ablates SVDD vs. plain diffusion sampling, showing improved complex scRMSD, conditional GT-RMSD, and ipTM when using reward-guided selection versus not using.
>   - We highlight this comparison in Table 4 and emphasize that SVDD is optional and improves performance without changing training in the paper revision.

---

> > ### Author Response · Authors · 2025-11-25
> > **Response to Reviewer 9N8E (II)**
> >
> > |Method|Success rate (%↑)|scRMSD (Å↓)|scTM-score (↑)|LDDT (↑)|qTMclust Diversity (↑)|
> > |-|-|-|-|-|-|
> > |w/o L_geom|92.77|4.94|0.69|0.71|1.00|
> > |w/o L_lj|95.30|4.67|0.69|0.72|0.94|
> > |w/o L_str|58.64|5.85|0.67|0.68|0.83|
> > |w/o L_rmsd|30.08|16.72|0.39|0.48|0.68|
> > |Full loss|97.38|3.43|0.71|0.74|1.00|
> >
> > - Auxiliary losses
> >   - Our current loss combines sequence cross-entropy, FAPE, RMSD, geometric penalties, and a Lennard–Jones clash term. To better show the effect of each component, we **add an ablation table** on single RNA co-design task comparing settings without certain components, and report performances metrics.
> >   - As shown in the table above, removing the global RMSD term has the most dramatic effect, confirming that $l_{rmsd}$ is critical for enforcing globally correct RNA folds rather than merely local improvements. Removing the structure-alignment loss also significantly hurts performance, indicating that the RF2NA-guided structural prior is essential for robust sequence–structure co-consistency and maintaining diverse yet correct conformations. In contrast, dropping the auxiliary geometric and Lennard–Jones terms yields more moderate degradation. These results support our design, with the rmsd and str terms providing the bone of global correctness, and the geometric and steric losses further sharpening physical plausibility and stability.
> >
> > |Method|Success rate (%↑)|scRMSD (Å↓)|scTM-score (↑)|LDDT (↑)|scSeqRec (%↑)|qTMclust Diversity (↑)|
> > |-|-|-|-|-|-|-|
> > |RF2NA-backbone + gRNAde (Backbone)|19.34|17.59|0.34|0.42|45.50|0.80|
> > |RiboDiff|97.38|3.43|0.71|0.74|48.57|1.00|
> >
> > - Two-step “backbone-only + gRNAde” setup is an interesting study. Comparing this variant baseline against RiboDiff will isolate the effect of joint co-diffusion of sequence and structure, using the same diffusion backbone and training data.
> >   - We have implemented this **“RF2NA-backbone + gRNAde” study**, where we adjust our diffusion design to train only a backbone structure diffusion and then apply gRNAde for inverse folding. We evaluate this pipeline on our single RNA co-design task, under the same train/test splits and metrics. Note that the scRMSD/scTM etc. is calculated only on all backbone atoms given the setting difference. This setup is similar to the RNA-FrameFlow baseline provided in response to W1. As shown in the table above, the results remain far behind RiboDiff’s performances. These results support our claim that joint sequence–structure co-diffusion provides stronger sequence–structure self-consistency.
> >
> > - These ablations, together with the existing ones and new baselines, provide a more systematic picture of how each component contributes to performance. We include these additions to the paper revision.
> >
> > > [Q1] Recent all atom structure prediction models unify proteins rna etc into a consistent framework instead of having specialised structure predictions methods like RF2NA. Do you think something similar is realistic/desirable in design?
> >
> > This is an insightful question. We discuss our perspective as below.
> > - A unified approach is increasingly feasible. Models including RF2NA already move toward a unified structural prior over proteins, RNA, DNA, ligands and complexes (as detailed in the RF2NA paper). It is natural to expect analogous unified generative models that can design multi-component systems (e.g., protein–RNA–DNA complexes) within a single framework.
> > - In this work, we focus on **RNA and RNA complex co-design under data scarcity**. RF2NA provides rich cross-molecular priors tailored to these domains, and our diffusion layers are explicitly designed to respect RNA’s discrete alphabet and SE(3) symmetries. A fully unified all-atom generative model would need to simultaneously cover very diverse chemistries and interfaces (proteins, nucleic acids, small molecules), **potentially diluting capacity for RNA** if not carefully architected.
> > - Conceptually, unified generative models are highly desirable for designing multi-modal assemblies (e.g., ribonucleoprotein machines, CRISPR complexes, RNA–ligand systems). Our work can be viewed as a step in this direction, since we already leverage a unified biomolecular predictor (RF2NA) and show that pretraining-guided diffusion is an effective recipe under data scarcity. A natural extension is to generalize RiboDiff’s dual discrete–continuous diffusion to multiple residue types and chains, effectively moving toward the unified design framework envisioned in the question.
> > - We include a discussion of this point in the paper revision, emphasizing that extending pretraining-guided diffusion to fully unified protein–RNA–DNA–ligand design is both realistic and a promising next step.
> >
> >
> > We sincerely thank you for your time! Hope we have addressed your concerns through practical efforts and shown the contributions and significance of our work. We look forward to your reply and further discussions, thanks!
> >
> > Sincerely,
> >
> > Authors

---

### Official Review · Reviewer_gdKq · 2025-11-01

**Soundness:** 3
**Presentation:** 3
**Contribution:** 3
**Rating:** 8
**Confidence:** 3

**Summary:**

This paper introduces a joint generative framework for RNA sequence and structure co-designs by using pretrained RoseTTAFold2NA as a denoiser for a dual-track diffusion model. They combine a discrete diffusion process for RNA sequence generation with a SE(3)-equivalent diffusion process for structure design, while leveraging the shared information embeded in the pretrained molecule folding model. They also introduced a RL techinquie to further optimize the generation during inference time.

In the empirical experiments, they showed their pretrained joint method consistently outperforms other joint baselines such as MMDiff, RNAFlow across three tasks settings, including single RNA co-design, conditional RNA designs.

**Strengths:**

1. The paper is well structured, and the motivation for leveraging pretrained cross-molecular priors is clearly articulated.

2. Integrating RoseTTAFold2NA into both discrete and continuous diffusion streams for RNA design is a clever and impactful idea that enables joint modeling of RNA sequence and structure, particularly given the scarcity of RNA data.

3. The method shows significant performance improvements over baselines across different metrics in single RNA design and significant gains in complex design.

4. The proposed framework is flexible and versatile; it naturally extends to conditional generation tasks and supports inference-time RL guidance.

**Weaknesses:**

1. Comparisons are primarily against a small set of joint models (MMDiff, RNAFlow, Random Generation). It would further strengthen the paper to compare against separately trained structure generators + inverse folding tools (gRNAde, RNAFrameFlow-style pipelines) or sequence generators + folding tools.

2. RoseTTAFold2NA is pretrained on large biomolecular datasets, including PDB structures that may overlap with evaluation sets; the paper should more clearly address dataset leakage or steps taken to avoid it

3. While the integration of RF2NA into diffusion is novel for RNA, the general paradigm mirrors prior work in protein design (RFdiffusion), slightly reducing novelty.

**Questions:**

Overall solid paper. Despite minor concerns about data leakage and missing comparisons with separate-stage baselines, the empirical gains and clear methodology justify acceptance.

---

> ### Author Response · Authors · 2025-11-25
> **Response to Reviewer gdKq**
>
> Dear Reviewer gdKq,
>
> Thank you for your appreciation of our work and insightful comments! We have made efforts to thoroughly improve our work accordingly and provide responses for each concern here. Please also check the revised paper and Appendices for additional experiments and clarifications.
>
> > Comparisons are primarily against a small set of joint models. It would further strengthen the paper to compare against separately trained structure generators+inverse folding tools (gRNAde, RNAFrameFlow-style pipelines) or sequence generators+folding tools.
>
> We thank the reviewer for emphasizing the importance of broader baselines.
> - Our current choice of baselines was guided by the goal of comparing between joint sequence–structure modeling. For single RNA and complex co-design, MMDiff is (to our knowledge) the only publicly available joint diffusion model for RNA and RNA–protein complexes, and is therefore the most direct baseline. We also compared with RiboGen in the Appendix. For protein-conditioned design, we compare against RNAFlow, which represents a joint pipeline that conditions on a protein.
> |Method|Success rate (%↑)|scRMSD (Å↓)|scTM-score (↑)|LDDT (↑)|scSeqRec (%↑)|qTMclust Diversity (↑)|
> |-|-|-|-|-|-|-|
> |Random generation|0.00±0.00|39.74±4.82|0.05±0.03|0.23±0.05|1.06±0.40|0.99±0.01|
> |MMDIFF|8.86±3.12|35.77±5.15|0.12±0.06|0.33±0.07|23.90±8.42|**1.00±0.00**|
> |RNA-FrameFlow + gRNAde (Backbone)|15.52±4.33|18.65±4.27|0.32±0.08|0.43±0.12|45.65±2.23|0.76±0.10|
> |RiboDiff|**97.38±4.86**|**3.43±0.51**|**0.71±0.04**|**0.74±0.06**|**48.57±4.20**|**1.00±0.00**|
>
> - That said, we agree that evaluating two-stage pipelines like structure generator + inverse folding will further clarify the value of joint co-design.
>   - We have added a baseline that first designs backbones with RNA-FrameFlow and then applies gRNAde to generate sequences for those backbones, closely mirroring the pipeline described in RNA-FrameFlow paper. To align with our evaluation pipeline for fair comparison, we use RF2NA for forward-folding instead of RhoFold to calculate the reference structure.
>   - For RNA-FrameFlow model, we follow their paper and use 6 IPA blocks with 3 torsion predictor layers, and $N_T = 50$. We adapt the architecture and implementation from their official code, and run training till convergence. We evaluate this pipeline on our single RNA co-design task, under the same train/test splits and metrics (scRMSD, scTM, lDDT, scSeqRec, qTMclust) as in Tables 1. Note that the scRMSD/scTM etc. is calculated only on all backbone atoms given the setting difference of RNA-FrameFlow. Also, scSeqRec under this setting is essentially an evaluation of gRNAde aligned to RF2NA in inverse folding.
>   - This directly address whether “backbone-then-sequence” pipelines can match the self-consistency and sequence-realizability of our joint co-design. As shown in the Table above, RiboDiff’s performances remain strong compared with this two-stage pipeline. These results support our claim that joint sequence–structure co-diffusion with a pretrained RF2NA prior provides stronger and more diverse sequence–structure self-consistency than a backbone-then-inverse-folding pipeline, particularly under structure metrics that involve refolding from sequence. We have added these baseline results in the revised paper and explicitly discuss their strengths and limitations.
>
> - We also appreciate the suggestion to include a sequence generator + folding tool baseline. We considered this setting carefully, but we believe it is a less informative and somewhat misleading baseline for our evaluation protocol.
>   - Current evaluation in this field already uses external structure predictors as a common oracle. All recent joint-generation and inverse-folding works (including gRNAde, MMDiff, RNAFlow, and related protein-design methods) evaluate generated molecules by folding them with a strong external predictor (RF2NA, Chai-1, Rosetta, etc.) and then computing scRMSD/lDDT/TM metrics on the folded structures. In other words, the structure predictor is part of the evaluation pipeline, not the generative model itself. Our work follows this established protocol.
>   - Thus a sequence generator + folding pipeline conflates the generation quality with oracle power. If we treat it as a competing method, we are no longer comparing generative models under the same evaluation oracle, instead we are comparing composite systems where RF2NA now acts as a powerful optimizer inside the baseline. This makes it difficult to attribute performance differences to the generative model versus the folding oracle.
>   - For these reasons, we view it as a different problem formulation, and not directly comparable to our joint sequence–structure models under the standard evaluation protocol. To keep the comparison focused and interpretable, we therefore prioritize joint/inverse-folding baselines such as MMDiff and RNAFlow/gRNAde-style pipelines. We have included this discussion in the revision.

---

> > ### Author Response · Authors · 2025-11-25
> > **Response to Reviewer gdKq (II)**
> >
> > > RoseTTAFold2NA is pretrained on large biomolecular datasets, including PDB structures that may overlap with evaluation sets; the paper should more clearly address dataset leakage or steps taken to avoid it
> >
> > We appreciate the reviewer raising this concern and agree it is important to be explicit about how we handle it.
> > - For the conditional generation task, we **already use an RF2NA pre-training split**. All complexes that RF2NA used as test during its original training are placed only in our test set, and RiboDiff is fine-tuned on the remaining complexes. Thus there is **no overlap** between the data used to fine-tune our generator and those we evaluate on in the conditional setting. We will clarify this more prominently in Section 5.1.
> > - For unconditional generation tasks, our goal is not to predict labels for held-out targets, but to learn **generic priors over the joint sequence–structure distribution** and then sample from it. In this regime, there is no target-specific supervision at evaluation time (we are not asked to recover a specific test structure), so the usual supervised notion of “test leakage” does not apply in the exact same way.
> >   - The evaluation focuses on self-consistency and diversity of generated samples (scRMSD/scTM/LDDT after refolding, qTMclust diversity), rather than accuracy on a fixed test set. Even if RF2NA has seen some of the PDB structures used to train RiboDiff, this only means we are using a stronger prior, which is the intended role of large-scale pretraining in modern generative modeling.
> >
> > | Methods| Random generation|MMDIFF|RNA-FrameFlow + gRNAde|RiboDiff|
> > |-|-|-|-|-|
> > |Novelty|100.0%|82.6%|55.8%|76.0%|
> >
> > - That said, a more relevant risk in the unconditional setting is **memorization**, that the model might simply reproduce training structures instead of learning a broad prior. To address this, we add experiments on another metric, **novelty**, where we compute compute the TM-score between all pairs of generated samples and structures in our training set, and if the highest align for a generated sample is <0.45, it is considered novel. As shown in the table above, we achieve reasonably high novelty, suggesting that the model is not simply memorizing the training structures.
> > - Finally, we discuss in the paper revision that, like most current structure-aware generative models, our method relies on a large, pretrained structural prior and therefore inherits any dataset overlap at that level. Also, our additional analyses above suggest that our improvements come from better joint modeling and co-diffusion, rather than trivial copying of pretraining examples.

---

> > > ### Author Response · Authors · 2025-11-25
> > > **Response to Reviewer gdKq (III)**
> > >
> > > > While the integration of RF2NA into diffusion is novel for RNA, the general paradigm mirrors prior work in protein design (RFdiffusion), slightly reducing novelty.
> > >
> > > We agree that RFdiffusion pioneered the “pretrained structure predictor → diffusion generator” paradigm for proteins. Our work is inspired by this paradigm, and we further extends to the RNA domains, which is substantially more data-limited and structurally distinct, and develops a joint sequence–all-atom structure diffusion framework. We would like to highlight several key differences.
> > > - **Joint discrete–continuous co-diffusion for RNA sequence + all-atom structure.** RFdiffusion primarily generates continuous backbone coordinates. In contrast, RiboDiff jointly diffuses over discrete sequences and all-atom SE(3)-equivariant frames, with a coupled objective that enforces sequence–structure co-optimisation throughout the trajectory (Sections 4.3–4.4). This enables us to more directly optimize self-consistency metrics that require both a realizable sequence and a physically plausible structure.
> > > - We extend the RF2NA architecture to **conditional generation of RNA binders for fixed proteins**, leveraging its three-track representation to encode protein–RNA interfaces (Section 4.5). To our knowledge, this is the first pretraining-guided diffusion model that performs protein-conditioned RNA binder co-design, and we show strong improvements over RNAFlow and conditional MMDiff (Table 3).
> > > - We integrate SVDD (soft value-based decoding) as an **RL-inspired inference scheme** tailored to RNA sequence–structure design, with rewards based on self-consistency and interface confidence (Section 4.6, Appendix). Table 4 shows that this improves complex scRMSD and ipTM without any retraining.
> > > - RFdiffusion operates purely on proteins using RoseTTAFold, whereas RiboDiff embeds RoseTTAFold2NA, which is trained on proteins, RNA, DNA, and their complexes. We therefore exploit cross-molecular priors that RFdiffusion does not study. Also, RNA’s conformational flexibility and sparse structural data make this extension experimentally non-trivial. Our results show that a pretraining-guided approach yields >10× lower scRMSD vs MMDiff on single RNA and large gains on complexes.
> > > We will clarify this relationship to RFdiffusion more explicitly in the paper revision, positioning our novelty as (1) the first RF2NA-based joint sequence–structure diffusion model for RNA, (2) explicit discrete-continuous co-diffusion, and (3) conditional RNA–protein co-design with RL-style inference.
> > >
> > >
> > > We sincerely thank you for your time! Hope we have addressed your concerns through practical efforts and shown the contributions and significance of our work. We look forward to your reply and further discussions, thanks!
> > >
> > > Sincerely,
> > >
> > > Authors

---

### Meta-Review · Area_Chair_FSKr · 2026-01-06

**Summary:**

In this study, the authors proposed a joint diffusion model for RNA sequence-structure co-design that addresses three RNA-centric conditional generation tasks. Technically, the main contributions of this work include 1) adapting existing DFM-Multiflow work in protein co-design to RNA-centric multi-task scenarios and 2) demonstrating the usefulness of strong pre-trained priors. Experiments verify the proposed method's potential to some extent.

The main concerns of reviewers include 1) the technical novelty of the proposed method, 2) the solidity of the experimental part, and 3) the advantage of the pre-trained priors compared to simpler conditions. In the rebuttal phase, the authors revised the paper and added more analytic experiments, highlighting the challenges of the focused tasks and the advantages of the proposed method.

Overall, I think the problem is interesting and challenging indeed. Although the technical novelty is not strong, this work makes a meaningful attempt to explore the potential of cutting-edge ML methods for the problem.

**Reviewer Concerns:**

I think most concerns have been resolved in the rebuttal phase.

**Reviewer Scores:**

The reviewers would have maintained their scores if they had discussed with each other.

---

### Decision · Program_Chairs · 2026-01-26

Accept (Poster)